

# Toward less subjective metrics for quantifying the shape and organization of clouds

Thomas D. DeWitt[1], Timothy J. Garrett[1], and Karlie N. Rees[1]

[1]Department of Atmospheric Sciences, University of Utah, 135 S 1460 E Rm 819, Salt Lake City, UT 84112, USA;

**Correspondence:** Timothy J. Garrett (tim.garrett@utah.edu)

**Abstract.**

As clouds sizes and shapes become better resolved by numerical climate models, objective metrics are required to evaluate whether simulations satisfactorily reflect observations. However, even the most recent cloud classification schemes rely on quite subjectively defined visual categories that lack any direct connection to the underlying physics. The fractal dimension of cloud fields has been used to provide a more objective footing. But, as we describe here, there are a wide range of largely unrecognized subtleties to such analyses that must be considered prior to obtaining meaningfully quantitative results. Methods are described for calculating two distinct types of fractal dimension: an individual fractal dimension $D_i$ representing the roughness of individual cloud edges, and an ensemble fractal dimension $D_e$ characterizing how cloud fields organize hierarchically across spatial scales. Both have the advantage that they can be linked to physical symmetry principles, but $D_e$ is argued to be better suited for observational validation of simulated collections of clouds, particularly when it is calculated using a straightforward correlation integral method. A remaining challenge is an observed sensitivity of calculated values of $D_e$ to subjective choices of the reflectivity threshold used to distinguish clouds from clear skies. We advocate that, in the interests of maximizing objectivity, future work should consider treating cloud ensembles as continuous reflectivity fields rather than collections of discrete objects.

## 1 Introduction

By resolving kilometer-scale processes, the next generation of climate models is expected to bring about a "quantum leap . . . in our ability to reliably predict climate" (Shukla et al., 2009). However, first it will need to be shown that the models can accurately reproduce observations of the current climate state. The challenge is that making such comparisons can be surprisingly difficult because there is no consensus on which metrics are best suited to constrain model performance (Janssens et al., 2021).

Given the climate is a physical system, any metric would ideally be something that can be linked quite directly to climate physics. Moreover, it should also be easily and accurately measurable using e.g. satellite observations, and for a sufficiently wide range of possible atmospheric states. Metrics of cloud geometries are an obvious candidate because they are readily observed from space and they are now resolvable by kilometer-scale climate models (termed Global Cloud Resolving Models or GCRMs). Also, they are a particularly challenging test: as clouds pass overhead, they can dramatically alter the local radiation budget (Stephens et al., 2015), and at the same time they are the aspect of the atmosphere that climate models struggle most to represent (Ceppi et al., 2017; Sherwood et al., 2020).

Faced with the challenge of quantifying clouds' radiative impact, scientists have generally tried to "divide and conquer", first by categorizing clouds and then identifying the physics unique to each category. The foundations of this approach can perhaps be traced to the early 19th century when Luke Howard took inspiration from the newly proposed Linnaean biological classification scheme. Using the only instrument available at the time – the human eye – Howard proposed the Latin nomenclature that still provides the dominant lens through which clouds are viewed by the atmospheric sciences community (Howard, 1803). But it is easily forgotten that Howard did not create such categories using some objective theoretical framework–rather, the categories originated from clouds' subjective appearance.

It is increasingly recognized that the properties of individual clouds are less important to the climate system than the collective impact of a cloud field (Müller and Hohenegger, 2020). Taking inspiration from Luke Howard, a new categorization system is gaining traction, this time for patterns of many clouds, using the names of sugar, gravel, fish, and flowers (Stevens et al., 2020). While differences in physical properties such as brightness temperature can be found between sugar, gravel, fish, and flowers (Bony et al., 2020), the category definitions themselves remain subjective in the first place.



At first glance, any subjectivity in classification could be eliminated by developing image processing algorithms that de-

terministically divide clouds into pre-defined categories that align with human intuition (Bony et al., 2020). However, such

objectivity is somewhat illusory given that the algorithm itself would be designed primarily to correspond with subjective

cloud definitions.

We are proposing here an alternative for classifying clouds that ties together cloud geometries with cloud physics. Very

generally, one of the most important physical principles is "symmetry" (Noether, 1918). In simple terms, a system displays

symmetry when it does not change if the system is transformed in some way (Feynman et al., 2015). As an example, the Navier-

Stokes equations possess symmetry in time because they can be used equally well to describe today's atmospheric motions as

those on some other day in the year 2100. Noether demonstrated that temporal symmetry implies energy conservation.

Less widely recognized are symmetries of spatial scale (Lovejoy and Schertzer, 2013). Namely, for some phenomenon $f$,

scaling symmetry exists when $f(x) \propto f(cx)$ where $c$ is a constant that "rescales" $x$ to some other spatial scale. An example

applying to clouds is the power-law distribution of cloud areas. Satellite observations show that the number of clouds $n$ scales

as their area $a$ through $n \propto a^{-\alpha}$ with a nearly constant exponent that spans at least 6 orders of magnitude (Wood and Field,

2011; DeWitt et al., 2024). The distribution respects a scaling symmetry because, if the scale under consideration – in this

example cloud area – were to be changed through multiplication by a constant $c$, then the number of clouds is rescaled by some

other constant $c^\alpha$. On Earth, due to the finite size of the planet and the finite size of cloud droplets, symmetry cannot apply to

an infinite range of scales. Nevertheless, for the range over which it applies, scaling offers an important simplification: cloud

geometries observed at any one scale can shed light on their geometries at (nearly) any other scale.

The aim here is to introduce simple and objective scaling metrics for defining the geometries of clouds and cloud fields.

We focus on cloud size and shape, and explore the applicability of fractal metrics from the standpoint of satellite observations

(Sect. 2). Our methods are also easily applied to climate model output. A particular appeal of quantifying fractal dimensions of

clouds and cloud fields is that the values can be shown to intimately reflect atmospheric dynamics – specifically, an anisotropy

between the horizontal and vertical dimensions of atmospheric turbulent flows (Rees et al., 2024).

Ascribing fractal properties to clouds is not new. But, as we show in Sect. 3 there are important pitfalls to fractal dimension

analyses that have been almost entirely overlooked in prior literature. Specifically, the fractal dimension of clouds is typically

determined from a relationship between individual cloud perimeter and area, as first introduced by Lovejoy (1982). The strict

mathematical definition of a fractal dimension, however, is defined by the relationship of an object's perimeter to the mea-

surement resolution (Mandelbrot, 1982). The justification for Lovejoy's method has its origin in a very brief derivation by

Mandelbrot (1982). We show here that the two approaches are equivalent provided certain very strict geometric conditions are

satisfied that do not apply well to natural clouds.

Section 4 introduces a novel fractal dimension that helps circumvent some of these challenges as well as being better suited

for characterizing cloud fields. For classifications of fields of clouds with varying sizes and shapes, this "ensemble" fractal

dimension may prove particularly useful for the validation of model performance or to compare meteorological and climate

states while reflecting the symmetries of spatial scale that underly the physics governing dynamic processes in the Earth's

atmosphere.

**2   Datasets**

To analyze the geometries of cloud fields, we use a calibrated optical reflectance product $R$ from MODIS that is sensitive

to visible wavelengths between $620\,\mathrm{nm}$ and $670\,\mathrm{nm}$ (Band 1). Each image or granule considered here covers a domain that

is roughly $1950\,\mathrm{km}$ wide by $2030\,\mathrm{km}$ long and was collected during January 2021 between 60°S and 60°N. "Cloud masks",



defined as binary images distinguishing cloudy from clear sky, are created by setting each pixel with $R$ greater than a set

threshold to cloudy and the rest to clear. A range of thresholds are considered between $R = 0.05$, permitting detection of

nearly all visible clouds, to $R = 0.3$, which allows only the most highly reflective convective cores to be detected (Fig. 1).

Higher thresholds were also tested (Appendix C) but they resulted in too few clouds for robust statistical analyses.

To reduce contamination from bright objects that are not clouds, images are only considered if they are at least 99.9% over

water and contain less than 10% sun glint (Ackerman et al., 1998). To ensure that the entire domain is visible in full daylight,

granules are omitted if any portion has a solar zenith angle larger than 70°. These criteria are chosen as a compromise between

a sufficiently large sample size and ensuring accurate measurements. The result is a total of 72 images.

Individual clouds are defined as contiguous groups of adjacent cloudy pixels. Diagonally positioned pixels are not considered

to be contiguous (Kuo et al., 1993). Pixels are not uniform in size because the distance between a cloud and the viewing satellite

can change, and also because each cloudy pixel is viewed at an angle. With appropriate adjustments for viewing geometry,

perimeters $p$ are calculated by summing contiguous pixel edge lengths along the boundary of each cloud, and areas $a$ by

summing the area of each cloudy pixel within the cloud. Contortions along cloud edges are unresolved for the very smallest

clouds (Christensen and Driver, 2021), so clouds with $a < 10\,\mathrm{km}^2$ are omitted from calculations of the fractal dimension of

individual clouds (Sect. 3.2), and clouds with $p < 10\,\mathrm{km}$ are omitted from perimeter distribution calculations (Sect. 4.3).

## 3    Determination of the individual fractal dimension $D_i$




**Figure 1.** Example cloud masks generated using a selection of thresholds in optical reflectance $R$, compared to a RGB image. The image shown was taken on Jan 1 2021 at 14:00 UTC, and is centered at approximately 29.1°N, 48.6°W.





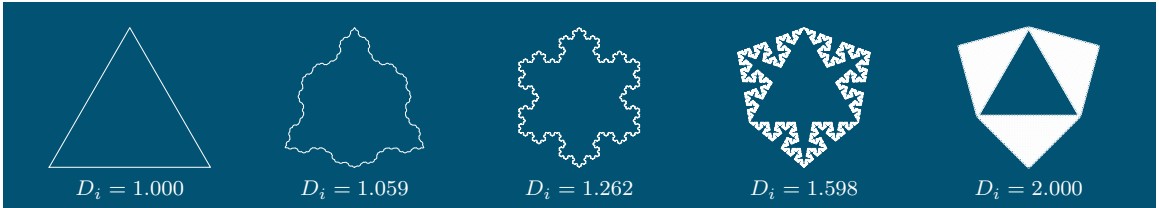

**Figure 2.** A generalized version of a self-similar Koch snowflake for varying fractal dimensions.

**3.1 Theory**

The original definition of the fractal dimension $D_i$ as proposed by Mandelbrot (1982) is that an individual cloud with perimeter

$p_i$ can be related to a variable "ruler length" or resolution $\xi$ through

$$p_i \propto \xi^{1-D_i}. \qquad (1)$$

Provided $D_i$ is not a function of $\xi$, the object can be termed to be "self-similar" because the measured quantity $p_i$ is a power

law function of the chosen spatial scale $\xi$ respecting a symmetry of spatial scale. Visually, self-similarity implies that the shape

of an object such as that shown in Fig. 2 repeats at many different spatial scales. Such exact repetition is not present in clouds,

but there is still a sense in which small-scale cloud edge contours appear similar to their larger-scale counterparts. To precisely

capture this tendency, we may modify the concept of self-similarity to only require that the *statistics* of the patterns – rather

than the exact pattern – to be similar across scales. This broader notion was termed "statistical self-similarity" by Mandelbrot

(1967). More specifically, statistical self-similarity exists when

$$\langle p_i \rangle \propto \xi^{1-D_i} \qquad (2)$$





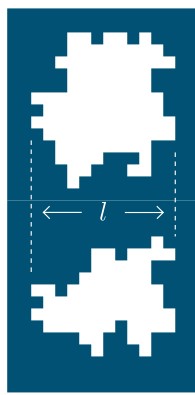

**Figure 3.** Two clouds may have different perimeters even if they are the same size $l$ (defined as bounding box width) and are measured at the same resolution $\xi$.

where the average is taken over some large collection of clouds with uniform size but not necessarily uniform perimeter (Fig. 3). The requirement of uniform size is intended to isolate changes in perimeter that are due solely to changes in the "roughness" of cloud edge, or the fractal dimension (Fig. 2; Imre (1992)). Size here is defined as a property that, unlike perimeter, is not

a function of resolution. It could be defined in a number of ways, but for simplicity of argument's sake (as discussed in more detail below) we start by defining size as the cloud bounding box width $l$ (Fig. 3).

It is difficult to apply Eqn. 2 directly to satellite images of cloud fields because clouds encompass a wide range of sizes and types, not to mention that satellite images are obtained at a fixed image resolution. It is for this reason that a different expression is usually used to determine the fractal dimension, one that relates satellite measures of individual cloud perimeters

$p_i$ to their areas $a_i$ (Lovejoy, 1982; Henderson-Sellers, 1986; Bazell and Desert, 1988; Gifford, 1989; Cahalan and Joseph, 1989; Jayanthi et al., 1990; Sengupta et al., 1990; Chatterjee et al., 1994; Gotoh and Fujii, 1998; Siebesma and Jonker, 2000; Luo and Liu, 2007; Peters et al., 2009; von Savigny et al., 2011; Brinkhoff et al., 2015; Batista-Tomás et al., 2016; Christensen





and Driver, 2021), namely

$$p_i \propto a_i^{D_i/2}. \tag{3}$$

In this case, $D_i$ is estimated from a simple linear fit to a scatterplot of the easily calculated quantities of $\log p_i$ and $\log \sqrt{a_i}$.

   Eqn. 3 for the fractal dimension is very different in form than the strict definition given by Eqn. 2. As justification, Lovejoy

(1982) referred to the work by Mandelbrot (1982) (p. 110), where it was proposed that the two expressions are interchangeable.

It is worth reviewing Mandelbrot's subtle argument in more detail because, as we will show, its assumptions do not strictly

apply to clouds.

On an intuitive level, any given cloud in a cloud field can be considered to have two independent geometric properties: size

and shape. By shape, we mean the roughness or complexity of the cloud boundary, which is perhaps the most visually notable

aspect that differentiates clouds, such as those shown in Fig. 1. To quantify cloud shape, suppose a subset of clouds within

the field, chosen such that each cloud has the same size $l_i$ but with varying perimeter $p_i$. This collection, represented by panel

(a) in Fig. 4, could be used to calculate the fractal dimension. One would "resample" each cloud by increasing $\xi$, creating a

collection of coarsened images as in panel (b). Such a resampling process would change each cloud's shape as finer contours

are no longer resolved in the coarsened images. Using a range of values for $\xi$ and the resulting range of perimeters $\langle p_i \rangle (\xi)$ at

each resolution, $D_i$ could then be obtained from a fit to Eqn. 2.

   Alternatively, the size could be varied as the shape stays the same. $D_i$ could be calculated from Eqn. 1 by taking a collection

of clouds having a range of sizes $l_i$, as in panel (c), and then resizing each cloud by normalizing the image width and height

dimensions by the cloud size, effectively zooming the image. Although all clouds were originally imaged with the same





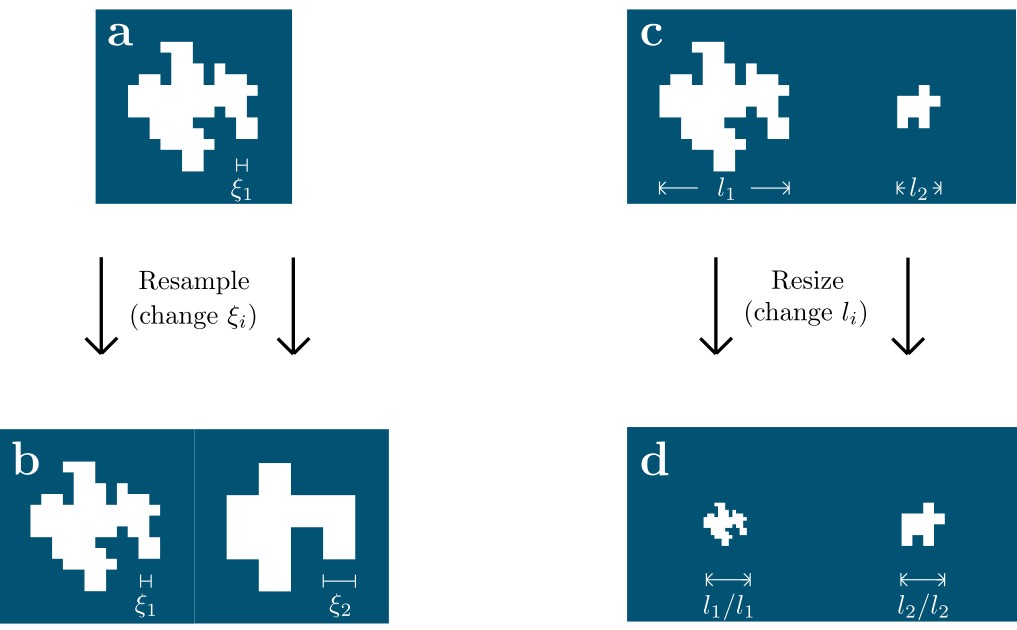

**Figure 4.** Self-similarity for a cloud field defined as an equivalence between the fractal dimension $D_i$ calculated from "resampled" (panels a and b) and "resized" (panels c and d) images of clouds. The right cloud in (b) was obtained from (a) by coarsening the resolution $\xi$ by a factor of 3, obtaining a collection of clouds (b) at varying resolutions as required by Eqn. 2. In contrast, the collection (d) was obtained by resizing two originally different clouds (c) such that their size $l_i$ is uniform. Self-similarity implies that fractal dimensions calculated from both (b) and (d) are the same.

resolution, the "larger" clouds, once resized, would appear to have a finer resolution than the smaller clouds. In fact, all length dimensions would change during resizing by the same factor $1/l_i$, including image width, cloud size, resolution, and perimeter. The resized clouds therefore have size $l_i/l_i = 1$, resolution $\xi_i/l_i$, and perimeter $p_i/l_i$. The fractal dimension $D_i$ could then be calculated by substituting the normalized perimeters and resolutions into Eqn. 1, leading to the expression $p_i/l_i \propto (\xi_i/l_i)^{1-D_i}$

or

$$p_i \propto \xi_i^{1-D_i} \, l_i^{D_i}. \tag{4}$$



The size metric $l_i$ must have dimensions of length given that it is used to normalize the length quantities $p_i$ and $\xi_i$.

Thus, there are two methods that can be used to calculate $D_i$ for a collection of clouds: resampling by varying shape and resizing by varying size. If the two methods yield the same value, then the clouds exhibit statistical self-similarity as defined

above. Note that, provided the field is self-similar, calculating $D_i$ by resizing has the advantage of enabling a more statistically robust fit. This is because any cloud in the field can be resized by normalizing by its size $l_i$, whatever its original size, and so the number of clouds that can be considered is much larger than if only resampled clouds of some fixed size are considered.

For the length dimension $l_i$, Mandelbrot (1982) proposed that the square root of the cloud area could be used, making the assumption that the cloud area is proportional to the area of the smallest bounding box $a_{\mathrm{BB}}$. If so, Eqn. 4 becomes Eqn. 3.

However, Mandelbrot's assumption that $a_i \propto a_{\mathrm{BB}}$ requires that cloud area is not a function of resolution because $a_{\mathrm{BB}}$ is not a function of resolution. If cloud area does in fact change with resolution, it would certainly be possible to calculate a statistical fit of $p_i$ to $a_i$ that yields a value of $D$ using Eqn. 3. But, this value of $D$ cannot be interpreted to be a fractal dimension defining the geometric properties of the cloud field (Imre, 1992), at least not one consistent with the original definition of the fractal dimension given by Eqn. 1. It is not even clear what the correct interpretation should be.

**3.2  Measurements**

Are Mandelbrot's assumptions valid for clouds such that Eqn. 3 is justified as a substitute for Eqn. 1? One potential issue is that clouds viewed from space have interior holes where the underlying surface can be viewed beneath. Resampling the cloud at a coarser resolution would cause the cloud holes to disappear so that $\sqrt{a}$ becomes a decreasing function of $\xi$. Cloud areas would not be self-similar, and so the perimeter-area relationship given by Eqn. 3 would not strictly hold. If a fit were calculated using

Eqn. 3 regardless, the calculated fractal dimension would be larger for large clouds that have more holes. Such an increase



has in fact been observed in several prior studies (Cahalan and Joseph, 1989; Gifford, 1989; Sengupta et al., 1990; Benner and

Curry, 1998).

To test whether the observed dependence of fractal dimension on cloud size is a real property of the clouds, or it is instead a

consequence of a false assumption that cloud area is resolution independent, any cloud holes could be filled to define a "filled

area" $a_f$ (von Savigny et al., 2011; Brinkhoff et al., 2015; Batista-Tomás et al., 2016). If instead Mandelbrot's assumption

$a \neq a(\xi)$ holds for clouds, then filling the holes should not be expected to affect the calculated fractal dimension. To illustrate,

consider the case where filling cloud holes increased every cloud's area by a constant factor $c$ such that $a_\mathrm{f} = ca$. This would

not affect $D_i$ because the coefficient $D_i/2$ relating $\log p$ to $\log a$ on a scatter plot of cloud sizes would remain unchanged,

consistent with the required property that the cloud field is scaling.

Figure 5 shows such a comparison between cloud perimeter and the square root of area, for filled and unfilled clouds

observed using MODIS as described in Sect. 2. In the imagery, filled areas $a_f$ and filled perimeters $p_f$ are calculated by first

identifying all contiguous clear regions that are not connected to the largest contiguous clear region in the image (the clear-sky

"background"), and then assigning those interior regions, or holes, as being cloudy. Then, $a_f$ and $p_f$ are calculated from the

hole-filled cloud mask in an equivalent manner as $a$ and $p$ as described in Sect. 2. To determine any size dependence of fractal

dimension calculated from Eqn. 3, regressions between $\log p$ and $\log \sqrt{a}$ are divided into four different decades in cloud area:

$10 \, \mathrm{km}^2$ to $10^2 \, \mathrm{km}^2$, $10^2 \, \mathrm{km}^2$ to $10^3 \, \mathrm{km}^2$, and $10^3 \, \mathrm{km}^2$ to $10^4 \, \mathrm{km}^2$.

Figure 5 shows that the "unfilled" fractal dimension has a higher value than the "filled" dimension at all scales (a similar

observation has been made of noctilucent clouds by Brinkhoff et al. (2015)). More importantly, the unfilled dimension increases

from $1.419 \pm 0.003$ for the smallest cloud size class to $1.8 \pm 0.2$ for the largest cloud size class. By contrast, the filled fractal

dimension does not display any statistically significant scale dependence.





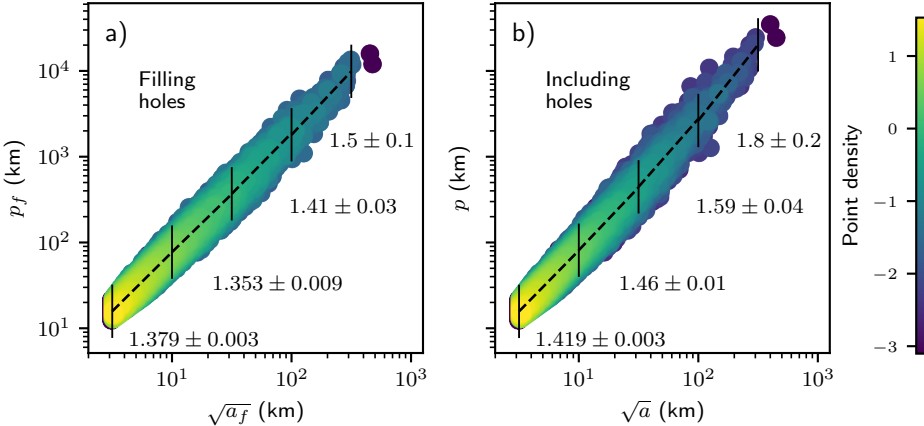

**Figure 5.** Dependence of the individual cloud fractal dimension on scale for clouds without holes (a) and with holes (b) for a threshold $R = 0.1$. The perimeters $p_f$ and areas $a_f$ are obtained from clouds after any holes are filled in, while $p$ and $a$ are calculated as described in Sect. 2. The listed fractal dimensions $D_i$ are calculated using regressions (dashed lines) to three different size ranges delineated by the vertical black lines.

For filled clouds, calculated values of $D_i$ change minimally with the reflectance threshold $R$ as shown in Fig. 6. For six different thresholds ranging from $R = 0.1$ to $R = 0.35$, $D_i$ only ranges from $1.374 \pm 0.001$ to $1.362 \pm 0.002$. Values of $D_i$ are consistent with values obtained in the minority of prior studies of cloud fractal properties that explicitly mentioned that cloud holes were filled (von Savigny et al., 2011; Batista-Tomás et al., 2016).

Summarizing, a fractal dimension $D_i$ defining a collection of clouds may be calculated from a perimeter-area relationship through Eqn. 3, in which case it offers a succinct metric for understanding cloud dynamics or for evaluating or comparing measurements and models. However, for the perimeter-area exponent to represent a true "dimension" mathematically, there is an a priori requirement that cloud areas not be a function of sampling resolution. This mathematical requirement can be satisfied, but only if cloud holes are filled, a procedure that might be argued to neglect an important physical property of the

cloud field.



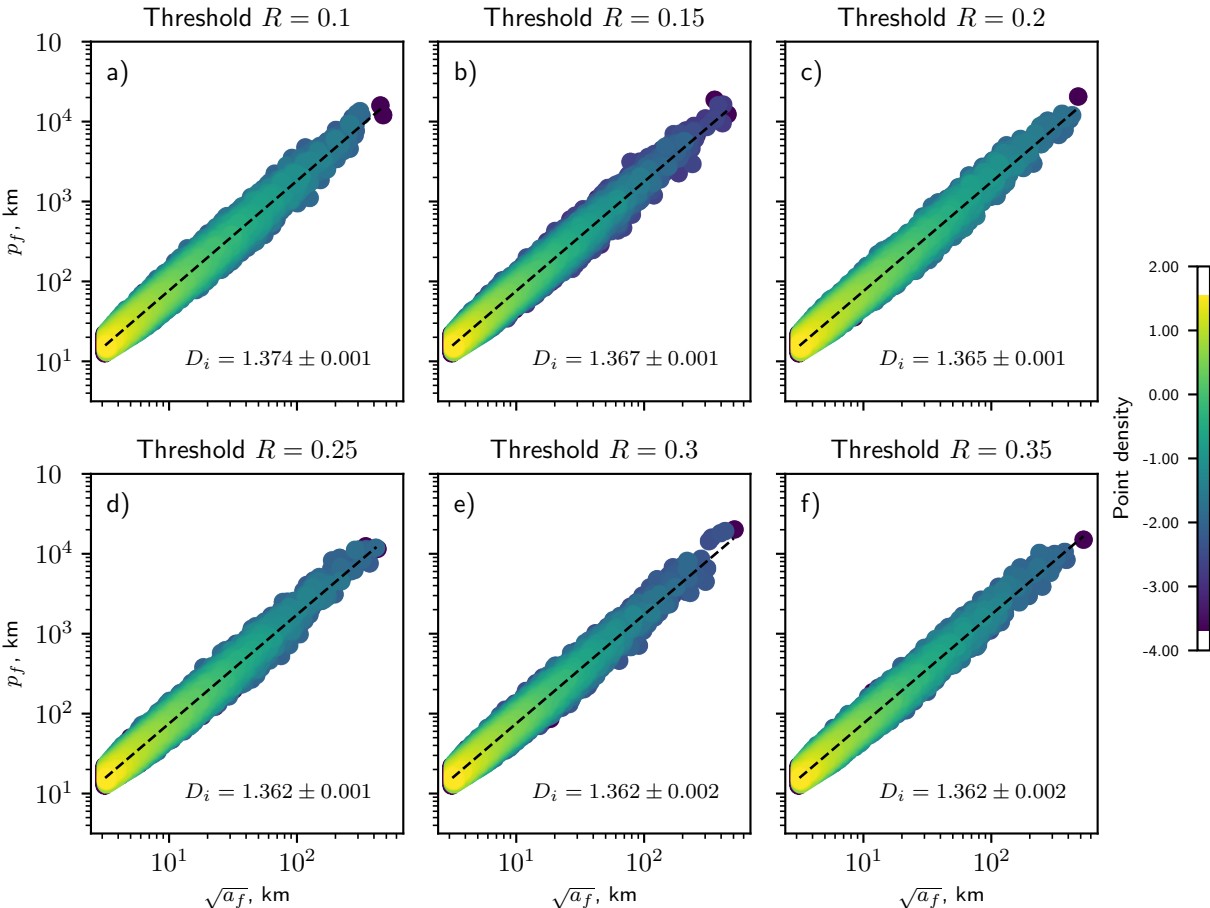

**Figure 6.** Calculations of the individual cloud fractal dimension $D_i$ for a range of cloud thresholds in reflectance $R$. Each plotted point represents one cloud and colors indicate logarithmically-scaled point density. Values $a_f$ and $p_f$ are the "filled" area and perimeter, that is, the perimeter and area of each cloud after cloud holes have been filled. Values for $D_i$ are obtained as a linear regression to $\log \sqrt{a_f}$ vs. $\log p_f$.

## 4 The ensemble fractal dimension

### 4.1 Theory

An alternative approach for defining the fractal properties of a cloud field according to Eqn. 2 is to employ what we term the

"ensemble fractal dimension" $D_e$. The mathematical definition is analogous to Eqn. 1, but with the mean cloud perimeter $\langle p_i \rangle$





being replaced by the total cloud perimeter $P$ :

$$P = \sum_i p_i \propto \xi^{1-D_e}. \tag{5}$$

This definition of the ensemble fractal dimension was employed by Rees et al. (2024) to demonstrate, using satellite obser-

vations, that the turbulent properties of the atmosphere are not 2D at large scales and 3D at smaller scales as is commonly

supposed (Nastrom et al., 1984). This relationship is derived analytically in Appendix A, where we show that $D_e$ captures two

orthogonal effects: individual cloud edge complexity and the size distribution of clouds in the field.

Curiously, where most past studies only measured $D_i$ (Lovejoy, 1982; Cahalan and Joseph, 1989; Siebesma and Jonker,

2000; Peters et al., 2009; Yamaguchi and Feingold, 2013; Christensen and Driver, 2021), some have in fact determined $D_e$

without discussion of how the two measures differ (e.g. Lovejoy et al., 1987; Carvalho and Dias, 1998; Janssens et al., 2021).

By considering the lengths of idealized island coastlines, Mandelbrot (1982) made the distinction clear: the ensemble fractal

dimension $D_e$ applies when "lumping all the islands' coastlines together."

For a better understanding of the numerical distinction between $D_i$ and $D_e$, consider again that cloud fields have the com-

bined properties of size and shape as shown in the idealized cloud field shown in Fig. 7. If the measurement resolution $\xi$ is

fixed, then from Eqn. A11 an increased value of $D_e$ corresponds to an increased value of $P$. Given that increasing the fractal

dimension of any given cloud increases its perimeter, and therefore the total $P$, we can see that $D_e$ increases if $D_i$ increases.

However, there is the added consideration of the relative numbers of small and large clouds – a subject now of considerable

interest for studies of convective cloud organization. To see how, consider re-drawing some cloud field such that the number

of small clouds increases while maintaining a fixed total cloud area, for example by dividing a few large clouds into a larger





number of smaller ones. Necessarily, this too would increase the total perimeter, since one would need to draw additional

cloud edges to divide up the cloud. Precisely how much $P$ changes depends on the change to the cloud size distribution. From

observations spanning a wide range of cloud sizes and climate states (DeWitt et al., 2024), a reasonable assumption is that the

distribution follows a power-law distribution with some exponent $\beta$ such that

$$n(p) \propto p^{-(1+\beta)} \tag{6}$$

where the total number of clouds $N$ is given by $N \propto \int_{p_{\min}}^{p_{\max}} n(p)dp$. In this case, a field with a large number of small clouds

relative to large clouds would have a larger value of $P$ and a higher value of $\beta$, assuming the total cloud area remains unchanged.

An example showing this dependence is shown along the ordinate of Fig. 7.

Thus, both shape and size affect $P$, numerically through $D_i$ and $\beta$. The ensemble fractal dimension must be a function of

both parameters. As a first guess:

$$D_e = \beta D_i. \tag{7}$$

In Appendix A, we propose based on analytical reasoning that Eqn. 7 is correct. Here, we take it as a conjecture to be tested

using satellite observations. If true, the ensemble fractal dimension given by Eqn. 7 provides a simple observational metric for

quantifying cloud field geometries, one that can also be tied to the invisible turbulent processes that roughen each individual

cloud's edge ($D_i$; Hentschel and Procaccia, 1984; Siebesma and Jonker, 2000) as well as the physics controlling the competition

for available energy among clouds of varying sizes ($\beta$; Garrett et al., 2018).







**Figure 7.** For a constant total cloud area, the total perimeter $P$ of a set of objects can change in two orthogonal ways: each individual object's perimeter can increase (horizontal; corresponding to a change in $D_i$) or the relative frequency of small objects can increase (vertical; corresponding to a change in $\beta$). The ensemble fractal dimension increases with both $\beta$ and $D_i$.



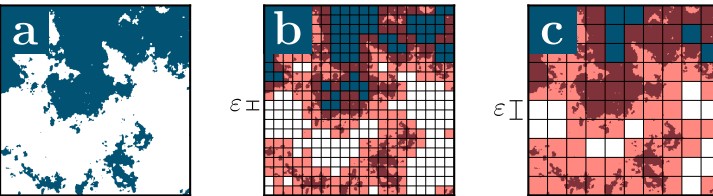

**Figure 8.** Calculation of the box dimension from a binary cloud mask (panel a). A grid with varying box size $\varepsilon$ is overlaid on the cloud mask (panels b and c) such that each box contains $\varepsilon \times \varepsilon$ image pixels. Here, $\varepsilon = 25$ for panel b and $\varepsilon = 50$ for panel c. The number of boxes required to cover all cloud edges (marked in red) are counted as a function of $\varepsilon$ to obtain $N(\varepsilon)$.

## 4.2 Calculating the ensemble fractal dimension in empirical data

To evaluate Eqn. 7, we employ two different methods to calculate the ensemble fractal dimension of cloud fields from satellite

imagery that are consistent with the definition of $D_e$ given by Eqn. 5. The most familiar approach is to use the "Minkowski-

Bouligand" or "box" dimension (Strogatz, 2018), which is calculated by first overlaying an evenly spaced grid on the cloud

mask and then counting the number $N$ of square boxes required to cover all cloud edges (red squares in Fig. 8). The procedure

is performed using a range of box (grid) sizes, where each box side length $\varepsilon$ covers some integer number of image pixels in

both directions. The fractal dimension calculated in this manner $D_{\text{box}}$ is defined by the relation

$$N(\varepsilon) \propto \varepsilon^{-D_{\text{box}}}. \tag{8}$$

Note that the box dimension is equal to the usual dimension for Euclidian objects; for example, a line has $D_{\text{box}} = 1$ and a disk

has $D_{\text{box}} = 2$ (Strogatz, 2018).

Although the actual resolution of the cloud mask $\xi$ is fixed, the box size $\varepsilon$ serves as an effective resolution for the purpose of

calculating the fractal dimension as defined by Mandelbrot, equivalent in the limit $\varepsilon \to 0$ to that definition given by Eqns. 1 or





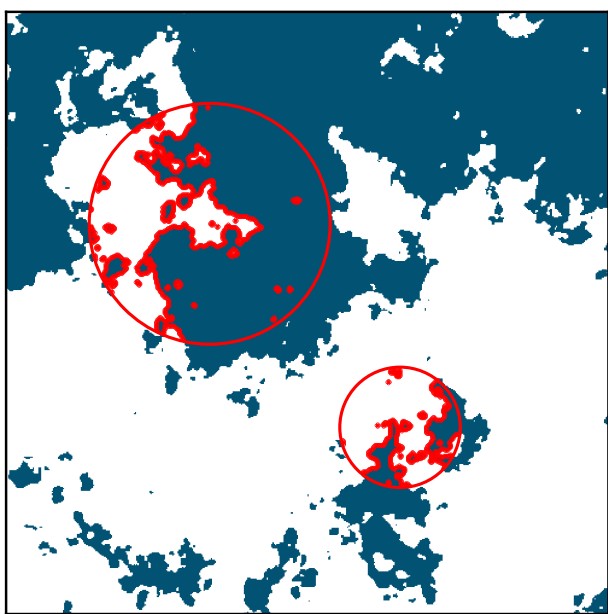

**Figure 9.** Example calculation of the correlation dimension. The red circles are centered at two randomly selected cloud edge pixels. For each circle center $j$, a component of the correlation integral $C_j$ is calculated by counting the number of cloud edge pixels (red) within the circle as a function of circle radius $r$.

5 since $P \sim \varepsilon N \sim \varepsilon^{1-D_{\text{box}}}$. There are, however, two other limiting cases. For a cloud field resolved at some finite resolution $\xi$,

the box dimension $D_{\text{box}}$ will tend to unity if the box length is smaller than the satellite resolution, that is $\varepsilon \lesssim \xi$. This is because

$\varepsilon$ will correspond to the Euclidian dimension of a line, in this case a pixel edge. Similarly, for boxes larger than the domain

$D_{\text{box}} = 0$ because a single box is always sufficient to cover all cloud edges, implying $N \neq N(\varepsilon)$ for such large boxes.

A second approach commonly used to measure the fractal dimension is the "correlation integral" method (Grassberger and

Procaccia, 1983). For clouds, this consists of first identifying all cloud edge pixels (i.e. cloudy pixels that are adjacent to a

clear pixel). For each pixel with index $j$, a circle of varying radius $r$ is drawn centered on that pixel (Fig. 9). The total number

of cloud edge pixels within the circle $C_j(r)$ is computed as a function of $r$. The fractal dimension is then calculated using the





"correlation integral" $C(r) = \sum_i C_i(r)$, which tends to be a power law of form

$C(r) \propto r^{D_c}$                                                                    (9)

where $D_c$ is the "correlation dimension". Although not strictly equivalent to Mandelbrot's ensemble fractal dimension (Eqn.

5), $D_c$ is empirically very close to $D_e$ (Strogatz, 2018), perhaps unsurprisingly given that both represent a relationship between

the scale under consideration (represented by $\varepsilon$ or $r$) and the density of cloud edge points (represented by $N(\varepsilon)$ or $C(r)$). In

practice, $C(r)$ follows a power law only for a finite range in $r$. As with the box dimension, the correlation dimension tends

toward the Euclidean dimensions of a single pixel for small $r$ and the domain shape for large $r$. Accordingly, below we will

only consider circles with $r \geq 3\xi$.

An important subtlety for calculation of $D_c$ is that, without proper care, even a few small circles might extend beyond the

domain boundary. If $C_j(r)$ is computed including such circles, $D_c$ will be biased towards the dimension of the artificially

straight domain boundary of $D_c = 1$. To remedy this issue, we only consider values of $r$ that are small in comparison to the

domain size and ensure that no circles extend beyond the domain boundary. Specifically, we enforce $r \leq r_{\max} = \min(L, W)/10$

where $W$ and $L$ are the domain width and length, respectively. We then only draw circles when the distance between the circle

center and the nearest domain edge point is at least $r_{\max}$. Note that even small circles should never be drawn closer to the

domain edge than $r_{\max}$, even if they did not extend beyond the domain edge. This is because a set of values for $C_j(r)$ that

contained data only from small circles would bias the sum $C(r)$ to be too large for small $r$.

There is a statistical advantage of using the correlation dimension for analysis of cloud fields, which is that $C(r)$ is created

using a much greater number of measured values compared with $N(\varepsilon)$. This is because $C(r)$ is a sum over many circle center



locations $j$ located along cloud edge – even for a single cloud. Accordingly, the correlation dimension is better suited for

satellite datasets with fewer clouds or those obtained at low resolution.

## 4.3 Measurements

For the box dimension, Fig. 10 shows the number of cloud edge boxes $N$ as a function of box length $\varepsilon$ for all 72 MODIS

images and six reflectance thresholds $R$ as described in Sect. 2. As expected, $D_{\text{box}} \approx 1$ for boxes that are small compared to

the satellite resolution, and $D_{\text{box}} = 0$ for very large boxes comparable to the domain size. For intermediate sized boxes, $D_{\text{box}}$

varies from $D_{\text{box}} = 1.68 \pm 0.05$ for threshold $R = 0.1$ to $D_{\text{box}} = 1.50 \pm 0.07$ for $R = 0.35$ suggesting (plausibly) that the more

reflective regions of clouds are some combination of either being more often large, with smaller $\beta$, or more Euclidean, with

smaller $D_i$.

For the correlation dimension, we find that observed values of $C(r)$ for clouds scale with $r$, implying a measured correlation

dimension for the cloud field ranging from $D_c = 1.74 \pm 0.01$ for a reflectance threshold of $R = 0.1$ to $D_c = 1.67 \pm 0.01$ for

$R = 0.35$ (Fig. 11). These values are somewhat higher than those obtained using the box dimension but display a similar trend

of decreasing values with increasing $R$.

To evaluate the hypothesis that $D_e = \beta D_i$ (Eqn. 7), $\beta$ is determined directly from the MODIS cloud masks. Power law

fits are obtained from a linear regression to a logarithmically binned and transformed histogram of cloud perimeter. Fits are

performed only for those bins for which at least 50% of clouds in that bin are entirely contained within the measurement

domain, ignoring the portion of the distribution that is dominated by large clouds that extend beyond the measurement domain.

The reason for this approach, as outlined in DeWitt and Garrett (2024), is that the perimeters of clouds that are truncated by the



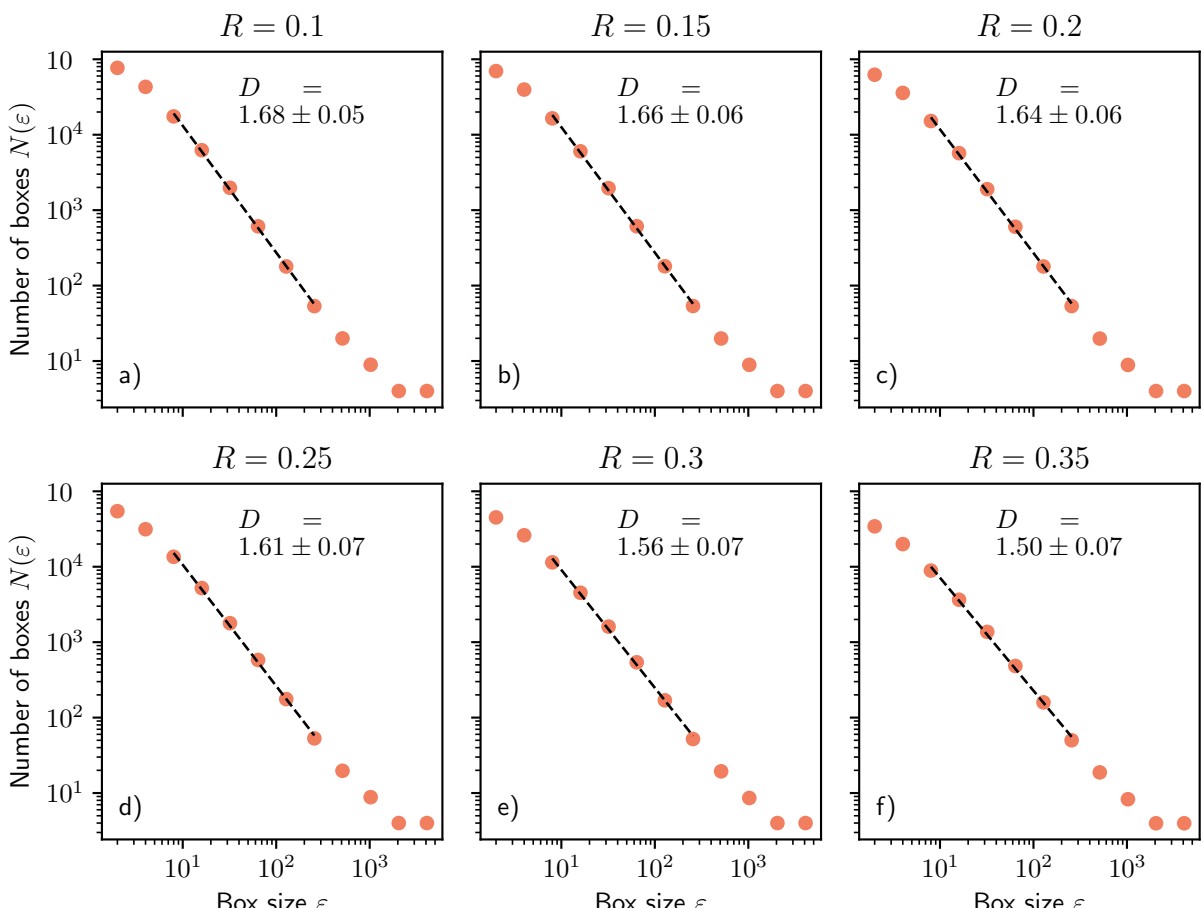

**Figure 10.** Measurements of the ensemble fractal dimension $D_e$ (Eqn. 5) using the box dimension $N(\varepsilon) \propto \varepsilon^{-D_{\text{box}}}$, where $N$ is the number of boxes of side length $\varepsilon$ required to cover all cloud perimeters (Fig. 8). At small and large $\varepsilon$, $D_{\text{box}}$ tends toward the values of a pixel ($D_{\text{box}} = 1$) or a point ($D_{\text{box}} = 0$), respectively. The box dimension attains a roughly constant intermediate value for medium-sized boxes, corresponding to the ensemble fractal dimension $D_e$.





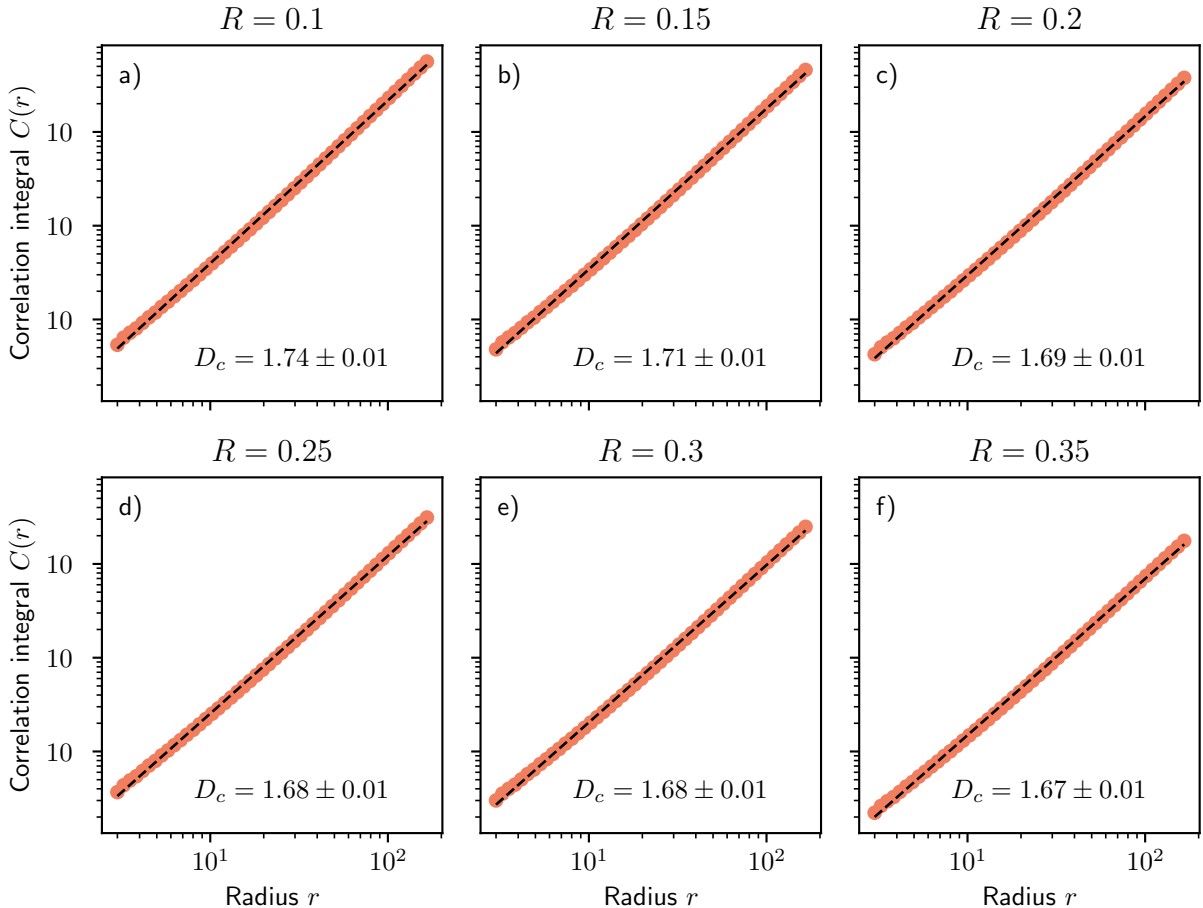

**Figure 11.** Measurements of the ensemble fractal dimension $D_e$ (Eqn. 5) using the correlation dimension, defined by $C(r) \propto r^{D_c}$, for the MODIS cloud ensemble for a selection of reflectivity thresholds $R$ defining cloud.

edge of a satellite image cannot be accurately measured. Such clouds tend to be large, and including them in the large-cloud

tail of the distribution can introduce a substantial bias in the calculated value for $\beta$.

Another consideration for the calculation of $\beta$ is the presence of "nested perimeters" (Appendix B), or the perimeters of

cloud holes and even nested clouds within the cloud holes. Here, we take the approach that each nested perimeter is in effect

a distinct cloud edge, for which there is a distinct value to be considered in the power law fit used to determine a value for

$\beta$ (Garrett et al., 2018). In effect, a single cloud may possess multiple perimeter values if it contains holes. The alternative





**Table 1.** Comparison of the individual fractal dimension $D_i$ with the three methods of calculating the ensemble fractal dimension $D_e$: the box dimension, correlation dimension, and the product $D_e = \beta D_i$ (Eqn. 7).

| Threshold | Measured $D_i$ | Measured $\beta$ | Product $D_i\beta$ | Correlation $D_e$ | Box $D_e$ |
|---|---|---|---|---|---|
| $R = 0.1$ | $1.374 \pm 0.001$ | $1.25 \pm 0.01$ | $1.72 \pm 0.02$ | $1.74 \pm 0.01$ | $1.68 \pm 0.05$ |
| $R = 0.15$ | $1.367 \pm 0.001$ | $1.27 \pm 0.02$ | $1.73 \pm 0.03$ | $1.71 \pm 0.01$ | $1.66 \pm 0.06$ |
| $R = 0.2$ | $1.365 \pm 0.001$ | $1.28 \pm 0.02$ | $1.74 \pm 0.03$ | $1.69 \pm 0.01$ | $1.64 \pm 0.06$ |
| $R = 0.25$ | $1.362 \pm 0.001$ | $1.32 \pm 0.03$ | $1.80 \pm 0.04$ | $1.68 \pm 0.01$ | $1.61 \pm 0.07$ |
| $R = 0.3$ | $1.362 \pm 0.002$ | $1.33 \pm 0.03$ | $1.81 \pm 0.04$ | $1.68 \pm 0.01$ | $1.56 \pm 0.07$ |
| $R = 0.35$ | $1.362 \pm 0.002$ | $1.32 \pm 0.02$ | $1.80 \pm 0.02$ | $1.67 \pm 0.01$ | $1.50 \pm 0.07$ |

approach would be to sum the exterior perimeter of a cloud with the perimeter of all the holes within the cloud prior to fitting.

Appendix B considers this distinction further and shows that the method employed here of including nested perimeters results

in values of $\beta$ that are larger by approximately 0.1.

Table 1 compares the calculated values of the box dimension and correlation dimension to the product $\beta D_i$. Generally, the

fractal dimension decreases with threshold $R$, although the dependence is significant only for $D_e$ with minimal sensitivity for

$D_i$ (see Appendix C for values of $D_e$ calculated using a wider range of thresholds). Such dependence is an indication that cloud

fields, are "multifractal," namely, the ensemble fractal dimension defining the field is a function of the thresholding scheme

used to binarize the cloud. Such multifractal behavior has been used to shed light on turbulent intermittency in the atmosphere

and indicates that the intensity of the turbulence varies spatially (Lovejoy and Schertzer, 1990).

All three methods for calculating the ensemble fractal dimension – the box dimension, correlation dimension, and $\beta D_i$ –

point to a value of $D_e \approx 1.7$, with a possible exception of the box dimension at higher reflectivity thresholds. Although $D_c$

and $D_{\text{box}}$ decrease with increasing threshold, the product $\beta D_i$ displays less of a trend due to the relatively weak dependence of

both $\beta$ or $D_i$ on $R$. More importantly, whichever method is used to calculate $D_e$, the values are substantially higher than those

for the individual fractal dimension $D_i$, which has a value of approximately $1.4$ (Fig. 6).

For cloud fields, this distinction between the two fractal dimensions $D_i$ and $D_e$ has been almost entirely overlooked by

previous studies, most of which only considered $D_i$ (Lovejoy, 1982; Henderson-Sellers, 1986; Bazell and Desert, 1988; Gif-

ford, 1989; Cahalan and Joseph, 1989; Jayanthi et al., 1990; Sengupta et al., 1990; Chatterjee et al., 1994; Gotoh and Fujii,

1998; Siebesma and Jonker, 2000; Luo and Liu, 2007; Peters et al., 2009; von Savigny et al., 2011; Brinkhoff et al., 2015;

Batista-Tomás et al., 2016; Christensen and Driver, 2021; Janssens et al., 2021; Cheraghalizadeh et al., 2024). The few studies

that measured $D_e$ did so by box-counting (Lovejoy et al., 1987; Carvalho and Dias, 1998; Janssens et al., 2021) but without

noting that $D_i$ and $D_e$ are different. Conflating the two different dimensions could lend a false impression of a discrepancy

between studies, when in reality the two values simply reflect the difference between the geometries of individual clouds and

those of a cloud field.

## 5   Conclusions

There is a need to validate numerical simulations of cloud sizes, shapes, and organization through comparisons with observa-

tions. Subjective classification schemes exist (Stevens et al., 2020), but objective mathematical metrics offer distinct advan-

tages, especially when they can be related directly to underlying physical processes such as symmetry principles and can be

applied uniformly across a wide range of spatial scales. Here, we explored the suitability of two metrics focused on the ge-

ometries of cloud edges: the individual fractal dimension $D_i$ that characterizes the geometric complexity of individual clouds,

and an ensemble fractal dimension $D_e$ that captures how the cloud field organizes hierarchically into structures spanning a

wide range of sizes and shapes. A Python package that implements the recommended methodology is made available (DeWitt,

2025).

Although both fractal dimensions have been previously studied, the distinction has not been widely discussed. The satellite observations of cloud fields we present here show a significant quantitative difference, with $D_i \approx 1.4$ and $D_e \approx 1.7$. We propose and observationally test that the two quantities can be related through the exponent $\beta$ for a power-law fit to the number distribution of cloud perimeters in the cloud field, namely $D_e = D_i\beta$.

By strict definition, the individual fractal dimension $D_i$ is determined from the resolution dependence of the perimeter of an individual cloud. Because satellites have fixed resolution, the more common technique is to consider a cloud ensemble and to fit the logarithm of cloud perimeter to the logarithm of cloud area. We show that this approach is fraught because clouds have holes. Not filling cloud holes violates the strict definition fractal dimension, while filling the holes misses an important physical property defining the cloud field.

Calculation of the ensemble fractal dimension $D_e$ bypasses these issues and offers a more objective alternative to sugar, gravel, fish, and flowers (Stevens et al., 2020), particularly if calculated as a correlation integral. The ease of its calculation of the correlation integral in satellite imagery makes it well-suited for evaluating the accuracy of atmospheric numerical simulations or for comparing regional meteorology. $D_e$ also has the potential to identify fundamental physical differences in atmospheric states relating to their scaling symmetries as it has been linked to the otherwise invisible and complex turbulent processes that shape cloud edge (Hentschel and Procaccia, 1984; Lovejoy et al., 1987; Siebesma and Jonker, 2000; Rees et al., 2024).

Although understanding clouds as objects might seem simple and intuitive at first, the hidden subtleties identified here prove surprisingly problematic, raising questions for future work. It is perhaps noteworthy that we tend to map a continuous reflectance field onto discrete entities called clouds, but that we do not do so for other continuous fields such as wind and



temperature. One might reasonably wonder whether the subtleties described here are worth the care required, or whether some

continuous field-based approaches might ultimately prove superior (Lovejoy and Schertzer, 2010).





**Appendix A: The relationship between the individual and ensemble fractal dimension**

Here we derive Eqns. 5 and 7, showing the link between the ensemble property $P$ and the individual cloud property $D_i$.

Conceptually, as illustrated in Fig. A1, there are two factors that lead to $P$ changing as the measurement resolution is varied.

The first relates to the fractal dimension for individual clouds $D_i$ (Eqn. 1). The second relates to the minimum resolved size of a

feature. This changes not the total number of clouds considered, but more strictly the number of *closed contours* defining cloud

edges. A closed contour could be defined by first starting at one cloud edge point and following the boundary between cloudy

and clear sky until the original point is reached. Any given cloud's boundary may be broken into multiple closed contours if

the cloud contains holes – one corresponding to its exterior edge and one for each hole.[1]

The relative contribution of small perimeters to the total is determined by the normalized size distribution of cloud perimeters, which from Eqn. 6 follows

$$n(p) \equiv \frac{dn}{dp} = n_0 p^{-(\beta+1)}, \qquad p_{\min} < p < p_{\max}, \qquad \beta > 0, \tag{A1}$$

$$n_0 = \frac{\beta}{p_{\min}^{-\beta} - p_{\max}^{-\beta}}, \tag{A2}$$

where $n_0$ is a normalization constant and $p_{\min}$ and $p_{\max}$ represent the smallest and largest measurable perimeters, respectively.

The total perimeter $P$ is the mean cloud perimeter $\int_{p_{\min}}^{p_{\max}} pn(p)dp$ multiplied by the total number of clouds $N$:

$$P = N \int_{p_{\min}}^{p_{\max}} pn(p)dp. \tag{A3}$$

---

[1]This can be expressed more precisely in the language of topology where, famously, a coffee cup is homeomorphic to doughnut, i.e. a coffee cup and a doughnut are "the same" topologically. To use technical language, consider a fractal curve with dimension strictly between 1 and 2 as a family of subspaces in the plane. Each subspace is defined by the curve of cumulative length $L$ that would be measured using a given measurement resolution $\xi$. If the family is homeomorphic, then $L \propto \xi^{1-D_i}$. If the family is not always homeomorphic, then $L \propto \xi^{1-D_e}$.





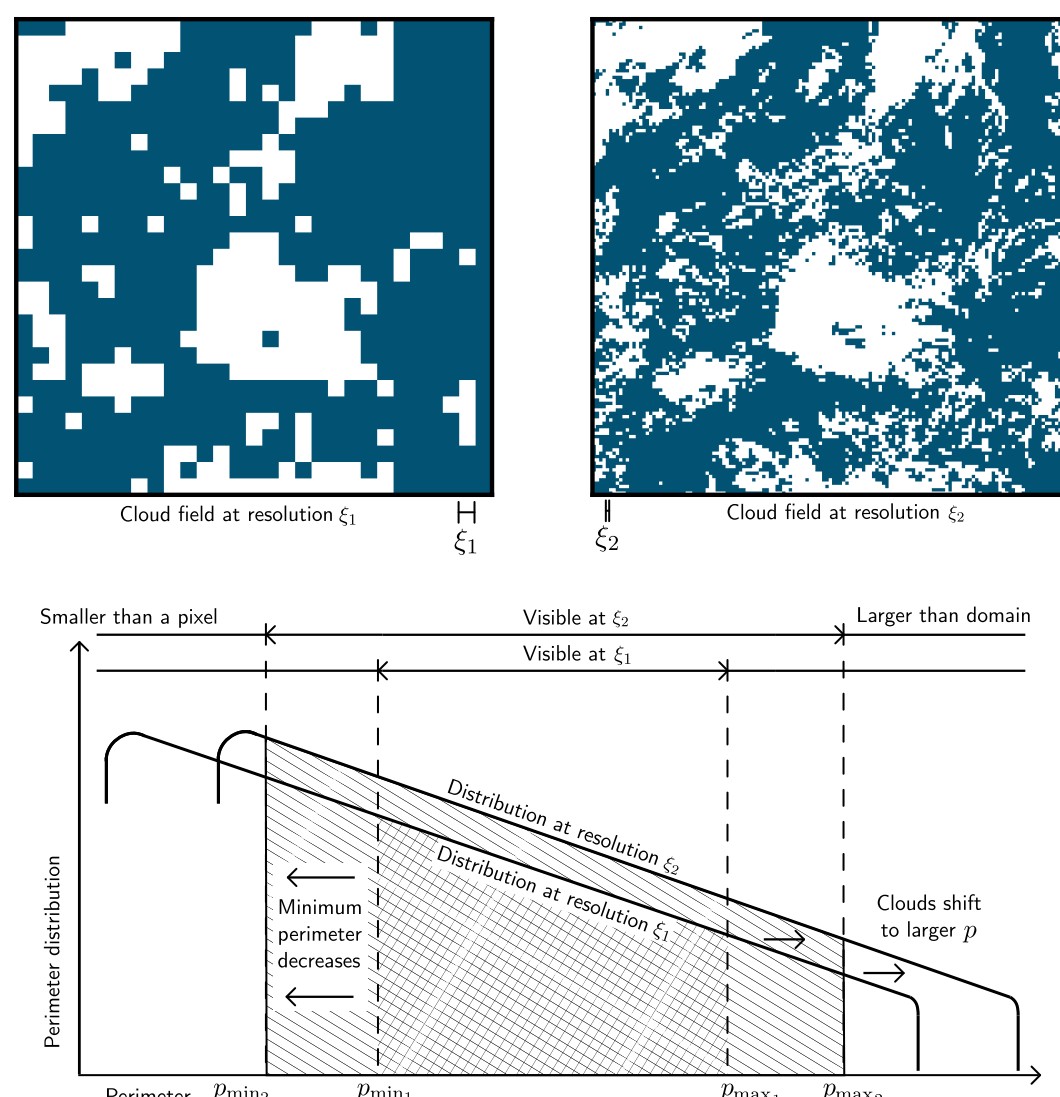

**Figure A1.** Cartoon of the changes to the logarithmically-scaled perimeter density (bottom) corresponding to a hypothetical rescaling from a resolution $\xi_1$ to a finer resolution $\xi_2$, i.e. $\xi_1 > \xi_2$ (top). The figure depicts the two effects to the total number of clouds as the resolution changes. The total number of clouds at $\xi_1$ (cross-hatched area) is smaller than the total number measured at $\xi_2$ (single-hatched plus cross-hatched area) for two reasons: first, the $p_{\min_2} < p_{\min_1}$, which adds the hatched area on the left. Second, the distribution for $\xi_1$ becomes, due to its rightward shift, vertically offset from the distribution at $\xi_2$, which adds the hatched area at the top. Note that, due to the logarithmic scale, the added area on the right caused by an increase in $p_{\max_2} > p_{\max_1}$ is negligible.





Satellite observations show $\beta \approx 1.26$ (DeWitt et al., 2024). Given that $\beta \neq 1$ the integral in Eqn. A3 may be calculated:

$$P = N \frac{\beta}{\beta - 1} \left( \frac{p_{\min}^{1-\beta} - p_{\max}^{1-\beta}}{p_{\min}^{-\beta} - p_{\max}^{-\beta}} \right), \qquad \beta \neq 1. \tag{A4}$$

We may now make several simplifications. The first is that the terms involving $p_{\max}$ may be dropped since $p_{\max}^{1-\beta} \ll p_{\min}^{1-\beta}$, as

previously observed to frequently hold (DeWitt et al., 2024). Additionally, we can replace $p_{\min}$ with the resolution $\xi$ times

some constant $c$. This is because the minimum possible measurable cloud size is determined by the resolution – for example,

if single-pixel clouds are counted towards the total perimeter then $p_{\min} = 4\xi$. With these simplifications,

$$P = N \frac{\beta c}{\beta - 1} \xi, \qquad \beta > 1, \quad p_{\max} \gg p_{\min}. \tag{A5}$$

The next step is to calculate the resolution dependence of $N$, the total number of clouds. As shown in Fig. A1, $N$ can be

conceptualized as the area under the curve of $n(p)$, and as the resolution changes, this area changes in two orthogonal ways:

first, the enclosed area grows or shrinks horizontally (the $p$ direction in Fig. A1) due to changes in $p_{\min}$, and second, the

distribution itself moves in $p$, $n(p)$-space as each cloud's perimeter coordinate changes via Eqn. 1, resulting in a vertical

change in the area under $n(p)$.

Expressed logarithmically, the two effects are, respectively,

$$\frac{d \ln N}{d \ln \xi} = \left. \frac{\partial \ln N}{\partial \ln \xi} \right|_{n(p)} + \left. \frac{\partial \ln N}{\partial \ln \xi} \right|_{p_{\min}}. \tag{A6}$$





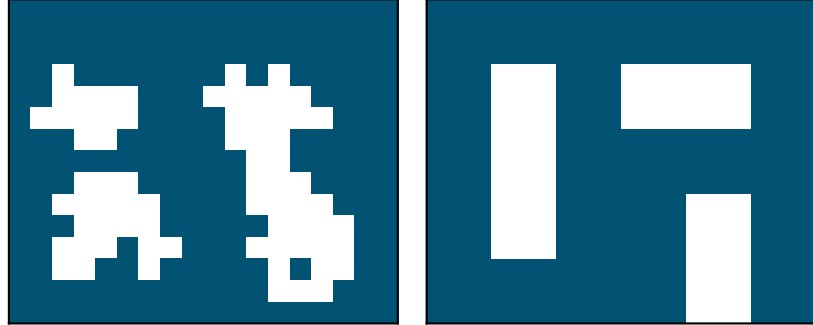

**Figure A2.** Conceptual example of how clouds could appear to break apart or combine as the resolution $\xi$ changes. Here, the right image is coarsened by averaging, for each pixel, the corresponding $3 \times 3$ pixel region in the left image and then rounding values to 1 (white) or 0 (blue). On the left side of the image, clouds appear to merge as the resolution is coarsened, while on the right, the opposite occurs. The ensemble fractal dimension is derived by neglecting the net effect of cloud merging or breaking apart.

For the first term, $n(p)$ is constant with respect to $\xi$ (Eqn. A1). Integrating Eqn. A1 from an arbitrary minimum perimeter $c\xi$ to $p_{\max}$ and again approximating $p_{\max} \to \infty$, we have $N \propto \xi^{-\beta}$ or

$$
\left. \frac{\partial \ln N}{\partial \ln \xi} \right|_{n(p)} = -\beta.
\tag{A7}
$$

The second term in Eqn. A6 represents the distribution itself shifting in perimeter space, and is in effect an "advection" equation analogous to a conservation law, but with "advection" here representing the change to each cloud's perimeter in perimeter space, not a change to the cloud's location in physical space. Expressed logarithmically, the advection equation is

$$
\left. \frac{\partial \ln N}{\partial \ln \xi} \right|_{p_{\min}} = -\frac{\partial \ln p}{\partial \ln \xi} \frac{\partial \ln N}{\partial \ln p}.
\tag{A8}
$$

Notably, Eqn. A8 assumes cloud number is locally conserved. As an example, Fig. A2 shows how this assumption could conceivably be violated: as the resolution changes, clouds could appear to combine or break apart, thereby changing the





number of clouds. We neglect these effects with the justification that any reduction in cloud number due to clouds combining

might easily be offset by clouds breaking apart (as in the example in Fig. A2), limiting any net change in cloud number.

From the integrated form of Eqn. A1, $N$ is proportional to $p^{-\beta}$, which implies that $\partial \ln N / \partial \ln p = -\beta$. Additionally, Eqn.

1 implies that $\partial \ln p / \partial \ln \xi = 1 - D_i$, so

$$\left. \frac{\partial \ln N}{\partial \ln \xi} \right|_{p_{\min}} = \beta - \beta D_i. \tag{A9}$$

Substituting Eqns. A9 and A7 into Eqn. A6 we obtain

$$\frac{d \ln N}{d \ln \xi} = -\beta D_i \tag{A10}$$

and, from Eqn. A5, the total perimeter $P$ is given by

$$P(\xi) \propto \xi^{1 - \beta D_i}, \qquad \beta > 1. \tag{A11}$$

If we define $D_e = \beta D_i$ as in Eqn. 7, we obtain Eqn. 5, consistent with Mandelbrot's assumption of a power law for $P$.

**The case of $\beta$ equaling unity**

While $\beta \approx 1.26 > 1$ has been observed to apply to satellite images of cloud perimeters that are seen from above (DeWitt et al.,

2024), the case of $\beta \approx 1$ is also physically realistic for the case that perimeters are measured within thin horizontal layers

(Garrett et al., 2018), as is possible in a numerical simulation.



In place of Eqn. A4, the total perimeter formula becomes

$$P = N p_{\min} \ln \left( \frac{p_{\max}}{p_{\min}} \right), \qquad \beta = 1. \tag{A12}$$

where, again, $p_{\min} = c\xi$ with $c$ a constant. In this case, the contribution of large clouds toward $P$ cannot be neglected. Instead,

we can use Eqn. 1 to obtain $p_{\max} = b\xi^{1-D_i}$ with $b$ some large constant that increases with domain area. From Eqns. A10 and

A12 we have

$$P \propto \xi^{1-D_i} \ln \left( \frac{b}{c\xi^{D_i}} \right), \qquad \beta = 1 \tag{A13}$$

In this case, $P(\xi)$ no longer exhibits a pure power law dependence on $\xi$ because there is an additional logarithmic factor.

However, if $b \gg c$, roughly corresponding to $p_{\max} \gg p_{\min}$, the logarithmic term is a slowly-varying function of $\xi$ for realistic

values of $p_{\max}$ and $p_{\min}$ determined using the MODIS observations described in Sect. 4.3 (not shown). In this case, $P(\xi)$ might

be approximated by the power law $P \propto \xi^{1-D_i}$ over a limited range of measurement resolutions. Conceivably, the equation

$D_e = \beta D_i$ (Eqn. 7) may still apply for $\beta = 1$ although further work is needed to determine if this is indeed the case.

**Appendix B: Comparison of the perimeter distribution exponent with and without cloud holes**

When calculating distributions for cloud perimeters, there is ambiguity about how to treat clouds with holes. On the one hand,

prior to calculating distribution parameters, the perimeter of any holes within a given cloud might be summed with the exterior

perimeter of the cloud, a method we term "summed values". Alternatively, each individual boundary might be considered as





**Table B1.** Comparison of calculated values of $\beta$ for two methods of treating cloud holes as a function of reflectance threshold $R$. Histograms from which values for $\beta$ are calculated are shown in Fig. B1. Values for $\beta$ are only calculated when the distribution spans two orders of magnitude (Stumpf and Porter, 2012) after bins containing less than 30 counts are removed (DeWitt and Garrett, 2024).

| Reflectance Threshold | $\beta$ for "nested values" | $\beta$ for "summed values" |
|---|---|---|
| $R = 0.05$ | - | $1.19 \pm 0.02$ |
| $R = 0.1$ | $1.25 \pm 0.01$ | $1.18 \pm 0.02$ |
| $R = 0.15$ | $1.27 \pm 0.02$ | $1.19 \pm 0.01$ |
| $R = 0.2$ | $1.28 \pm 0.02$ | $1.20 \pm 0.02$ |
| $R = 0.25$ | $1.32 \pm 0.03$ | $1.24 \pm 0.02$ |
| $R = 0.3$ | $1.33 \pm 0.03$ | $1.25 \pm 0.02$ |
| $R = 0.35$ | $1.32 \pm 0.02$ | $1.26 \pm 0.02$ |
| $R = 0.4$ | $1.35 \pm 0.02$ | $1.28 \pm 0.02$ |
| $R = 0.5$ | - | - |
| $R = 0.6$ | - | - |
| $R = 0.7$ | - | - |
| $R = 0.8$ | - | - |
| $R = 0.9$ | - | - |

contributing a unique perimeter value to the histogram from which the distribution exponent $\beta$ is calculated. This method, which we term "nested values", was used in Sect. 4.3. For example, a doughnut-shaped cloud would have a single value for its perimeter for the summed case but two values for the nested case, one corresponding to the hole's perimeter and one to the exterior perimeter.

Table B1 compares the sensitivity of calculations of $\beta$ (Eqn. 6) to whether the summed or nested values approach is taken. For all considered thresholds in reflectivity $R$, $\beta$ is larger if the nested value method is used. This is for two reasons. First, adding hole perimeter to a cloud's exterior perimeter increases that cloud's perimeter, placing it in a larger bin. Second, summing reduces the number of perimeters in a smaller bin because the hole perimeters are no longer counted in the smaller bin. Both effects act to decrease the slope of the distribution, making $\beta$ smaller, even as both cases produce size distributions that are

well-described by a power law distribution (Fig. B1).

Given that cloud holes do exist, and that they are not considered for $D_i$, we employ the "nested" methodology in the main text as it more realistically represents the role of cloud holes.





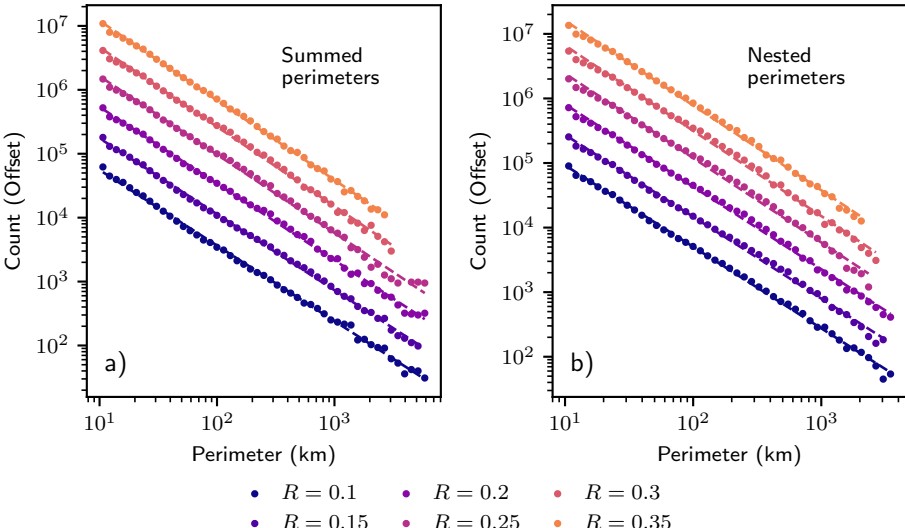

**Figure B1.** Histogram of cloud perimeters for various thresholds in reflectance $R$, calculated for both cases described in the text. Counts for thresholds larger than $R = 0.1$ are vertically offset by factors of 3 for clarity. Values and uncertainties for the slopes, which correspond to $-\beta$ (Eqn. A1), are listed in Table B1. Perimeter values smaller than $10\,\mathrm{km}$, bins with counts smaller than 30, or bins in which a majority of clouds extend beyond the measurement domain are omitted from the plot and regression (DeWitt and Garrett, 2024).

## Appendix C: Parameters for a wider range of reflectance thresholds

In the main text, reflectance thresholds $R$ used to define cloud were limited to a range between $R = 0.1$ to $R = 0.35$. The

reason is that higher thresholds reduce the number of clouds and therefore the statistical robustness of the results. Additional

thresholds, at which only some parameters can be reliably estimated, are listed in Table C1. Figures C1, C2, and C3 are as in

Figs. 6, 10, and 11 but for these additional reflectance thresholds.





**Table C1.** As in Table 1, but for a wider range of reflectance thresholds $R$. Values for $\beta$ are only calculated when the distribution spans two orders of magnitude (Stumpf and Porter, 2012) after bins containing less than 30 counts are removed (DeWitt and Garrett, 2024). Likewise, values for $D_i$ are only listed when $\sqrt{a}$ spans at least two orders of magnitude, and the box and correlation dimensions are only calculated when $C(r) > 30$ and $N(\varepsilon) > 30$ for all $r$ and $\varepsilon$, respectively.

| Threshold | Measured $D_i$ | Measured $\beta$ | Product $D_i\beta$ | Correlation $D_e$ | Box $D_e$ |
|---|---|---|---|---|---|
| $R = 0.05$ | $1.384 \pm 0.002$ | - | - | $1.77 \pm 0.01$ | $1.69 \pm 0.04$ |
| $R = 0.1$ | $1.374 \pm 0.001$ | $1.25 \pm 0.01$ | $1.72 \pm 0.02$ | $1.74 \pm 0.01$ | $1.68 \pm 0.05$ |
| $R = 0.15$ | $1.367 \pm 0.001$ | $1.27 \pm 0.02$ | $1.73 \pm 0.03$ | $1.71 \pm 0.01$ | $1.66 \pm 0.06$ |
| $R = 0.2$ | $1.365 \pm 0.001$ | $1.28 \pm 0.02$ | $1.74 \pm 0.03$ | $1.69 \pm 0.01$ | $1.64 \pm 0.06$ |
| $R = 0.25$ | $1.362 \pm 0.001$ | $1.32 \pm 0.03$ | $1.80 \pm 0.04$ | $1.68 \pm 0.01$ | $1.61 \pm 0.07$ |
| $R = 0.3$ | $1.362 \pm 0.002$ | $1.33 \pm 0.03$ | $1.81 \pm 0.04$ | $1.68 \pm 0.01$ | $1.56 \pm 0.07$ |
| $R = 0.35$ | $1.362 \pm 0.002$ | $1.32 \pm 0.02$ | $1.80 \pm 0.02$ | $1.67 \pm 0.01$ | $1.50 \pm 0.07$ |
| $R = 0.4$ | $1.365 \pm 0.002$ | $1.35 \pm 0.02$ | $1.84 \pm 0.03$ | $1.64 \pm 0.01$ | $1.41 \pm 0.08$ |
| $R = 0.5$ | $1.364 \pm 0.004$ | - | - | $1.473 \pm 0.006$ | $1.13 \pm 0.09$ |
| $R = 0.6$ | $1.37 \pm 0.01$ | - | - | $1.266 \pm 0.004$ | - |
| $R = 0.7$ | - | - | - | $0.881 \pm 0.007$ | - |
| $R = 0.8$ | - | - | - | $0.595 \pm 0.005$ | - |
| $R = 0.9$ | - | - | - | $0.16 \pm 0.01$ | - |

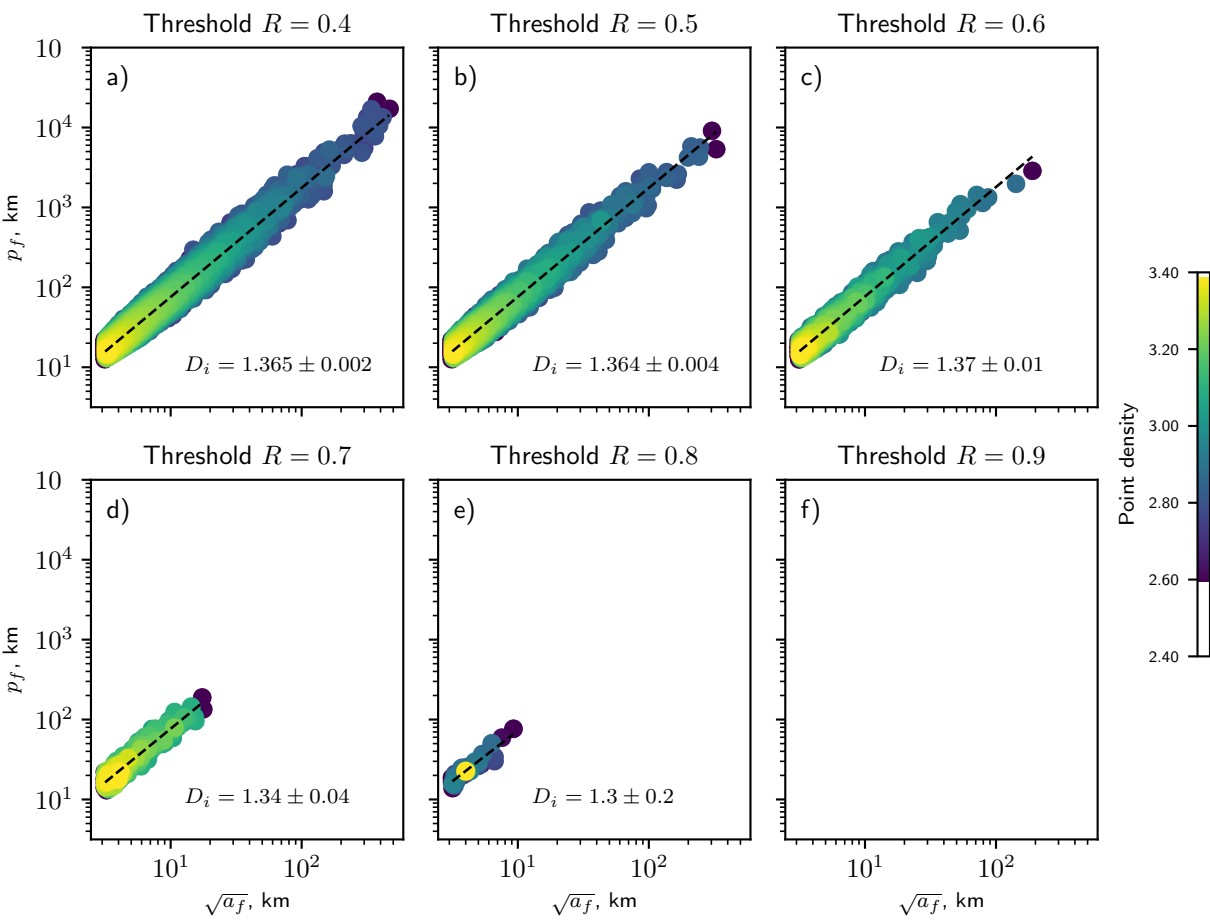

**Figure C1.** As in Fig. 6, but for larger reflectance thresholds $R$. Threshold $R = 0.9$ did not identify any clouds.





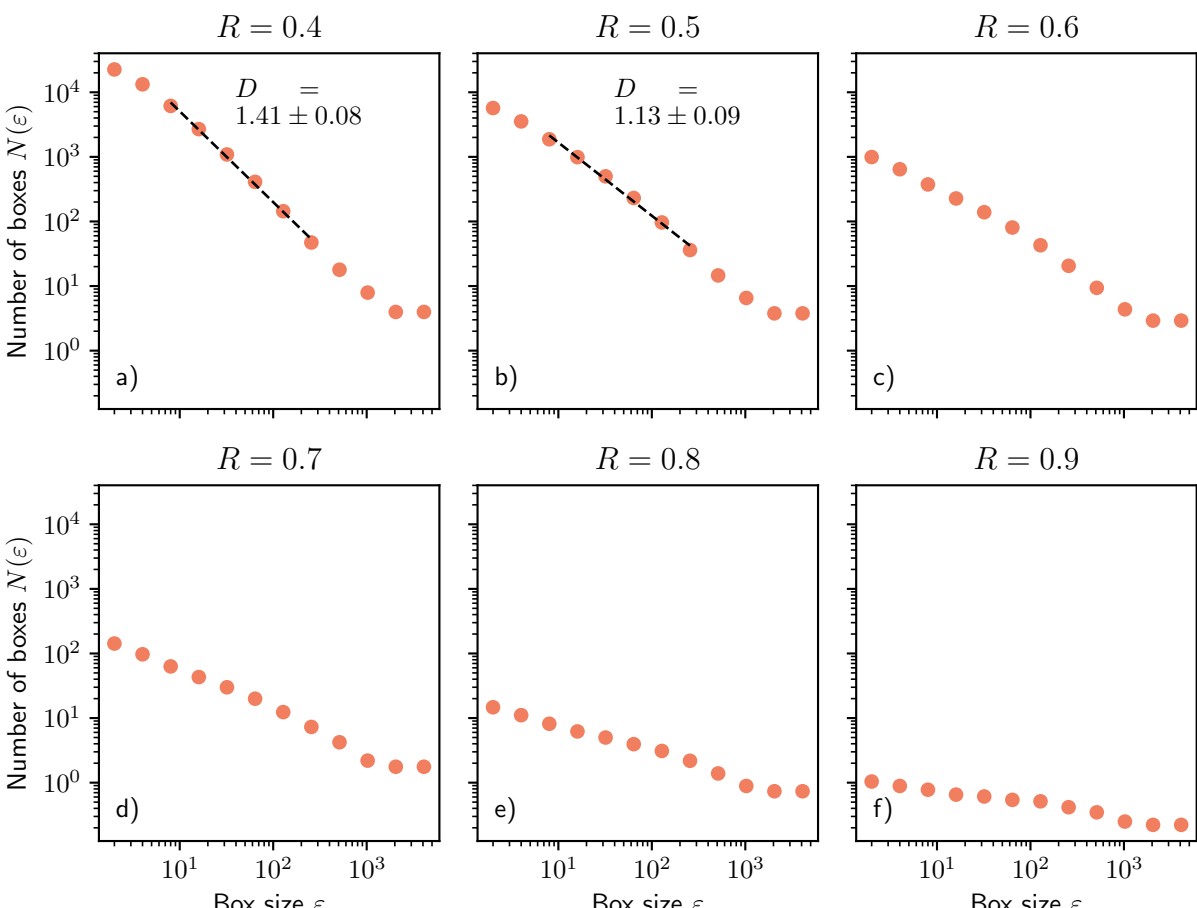

**Figure C2.** As in Fig. 10, but for larger reflectance thresholds $R$.





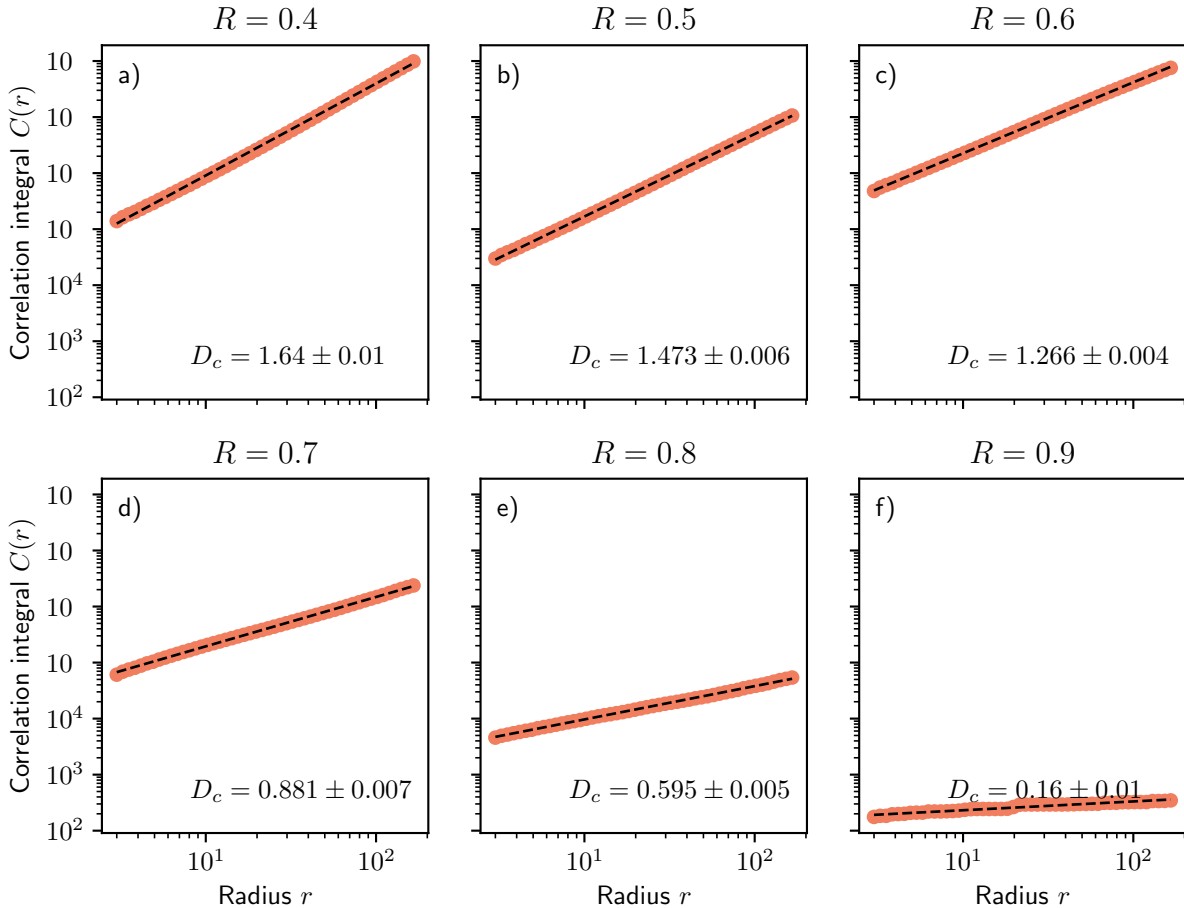

**Figure C3.** As in Fig. 11, but for larger reflectance thresholds $R$.



*Code and data availability.* The methodology recommended here has been implemented in a fully documented Python package named

`objscale`, available via `pip` (DeWitt, 2025). Additional code and MODIS data used for the analysis presented here is available at

https://doi.org/10.5281/zenodo.15844057.

*Author contributions.* TDD: conceptualization, formal analysis, sofware development, methodology and writing (original draft preparation).

TJG: conceptualization, funding acquisition, supervision, methodology and writing (review and editing).

*Competing interests.* At least one of the (co-)authors is a member of the editorial board of Atmospheric Chemistry and Physics.

*Financial support.* This research has been supported by the National Science Foundation (grant no. PDM-2210179).

*Acknowledgements.* Michael Kopreski of the University of Utah Dept. of Mathematics helped clarify the topological distinction between the

two fractal dimensions.



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
