# Peer review of "Toward less subjective metrics for quantifying the shape and organization of clouds"

_EGUsphere, 2025_

## Referee Comment (RC1)

**REVIEW OF EGUSPHERE-2025-3486**

Toward less subjective metrics for quantifying the shape and organization of clouds Anonymous Reviewer

**Summary**

This paper examines the use of two different metrics utilizing the scale-invariant nature of clouds as a quantifiable measure of the geometric characteristics of large-scale cloud fields based on satellite observations. Of special interest here is the distinction between the individual and ensemble fractal dimensions, and the latter is found to be a more robust representation of the geometric characteristics of the cloud field.

I found this paper to be interesting and scientifically meaningful. I do have a few issues with technical details and some of the assumptions made in this paper, but overall I believe that the work needs to be considered positively.

**Comments**

**L23**

The use of fractal dimensions to compare satellite observations to numerical simulations comes up a number of times in this paper, but the authors seem to conflate different scales and resolutions. Here, the authors mention kilometre-scale climate models; is the argument that satellite images (at 250-m resolution with MODIS, for example), can be used as a link "quite directly to climate physics" of these climate models? How?

**L26**

The two papers cited here are pretty clear about the measure of uncertainty, so the authors could be more clear about the struggle to accurately represent the radiative effects of clouds. Also, there have been quite a number of studies recently on this very topic since 2020.

**L87**

It would be nice to see the actual resolution, and exactly how much it varies.

**L120**

A "simple linear fit" can mean two things. Either that the (unwieldy) distribution has been estimated by a linear function, or that the power-law fit worked exceptionally well. Can we have a figure that actually shows the linear fit?

**L214**

It is indeed true that the perimeter and the size of a cloud are positively correlated, but the relationship is not linear (*cf.* https://doi.org/10.5194/acp-13-7795-2013). Either this significantly affects the power-law relationship, or I think it should be made clear that the power-law fit is not an exact representation of the cloud size distribution.

**L225**

I believe the authors are being intentionally ambiguous here, as the cloud field geometry being "tied to" turbulent processes can be used to mean anything; if the authors are implying that the ensemble fractal dimension  $D_{\rm e}$  from satellite images can be used to infer the strength of turbulent mixing processes, or thermodynamic properties, then I would love to see more details.

**L280**

There is very little discussion on how the linear fits in Figure 10 were obtained. The power-law distributions on the log-log plots are only marginally linear, and they are dependent on the choice of the bins and other factors (see the comment below); one could also argue that the *intermediate* sizes used for the linear fit have been cherry-picked. There is a lot of literature on using the power-law fit for the cloud size distribution, and I believe there is much to be justified here.

**L281**

This sounds like a significant amount of large clouds being filtered out. I do understand the need to filter truncated clouds, but given that larger clouds naturally have higher chances of extending beyond the observed domain, it sounds like a considerable chunk of the cloud field that is being omitted. For a typical MODIS scene, how much of the observed cloud field has been filtered out at this stage?

The authors mention that including those clouds will result in "a substantial bias", but could it be the other way around? Also, why the bins that include 50% of whole clouds? These choices seem too arbitrary to me.

Ultimately, I think this directly opposes the main goal of this paper, which is to come up with a measure of geometric features of a cloud field, especially for convective organization.

**Figure 10**

The exponents are missing on the y-axis (same for Figure 11).

I am not sure if I agree with the observation. The slope  $D_{\rm box}$  does tend towards 0 for small and large values of  $\varepsilon$  (for very different reasons), but I am not sure if we can say it tends towards 1, unless the authors mean that the slope is close to 1 for small  $\varepsilon$ , in which case I disagree.

**L300**

I do not see how this number came about; is 1.7 just an average across all estimates of the ensemble fractal dimension? I feel like these numbers depend more on the filtering of large clouds and the linear fit than the actual properties of the cloud field, as they behave differently at smaller thresholds.

**L330**

I believe it should be "strict definition of the fractal dimension".

**L332**

See my first comment.

---

## Author Comment (AC1)

We thank both reviewers for their helpful comments.

In this document, italics denote reviewer comments and the grey boxes contain excerpts of the manuscript where changes were made. Blue indicates added text and  in keeping with the changes file.

**Response to Reviewer 1's Comments**

*"L23: The use of fractal dimensions to compare satellite observations to numerical simulations comes up a number of times in this paper, but the authors seem to conflate different scales and resolutions. Here, the authors mention kilometre-scale climate models; is the argument that satellite images (at 250-m resolution with MODIS, for example), can be used as a link "quite directly to climate physics" of these climate models? How?"*

One of the main appeals of the metrics we propose is that they offer a resolution-independent way to quantify the statistics, as they quantify relationships *between* scales rather than *at* any given scale. We have added a sentence emphasizing this benefit:

> "ensemble" fractal dimension may prove particularly useful for  characterizing cloud fields as it quantifies relationships between clouds of different sizes in a way that the more common "individual" fractal dimension does not.
> We also aim to examine and improve the methodologies by which the fractal dimensions are calculated in order to give future model intercomparison or cloud classification studies better tools. A particular appeal of fractal metrics is that the values characterize statistical relationships *between* scales rather than statistics *at* any particular scale, for example enabling more straightforward comparisons between observations or models with differing resolutions. But for such benefits to be realized, the metrics must be accurately calculated in the first place such that their values do not depend on the particulars of domain size or dataset resolution. When accurately measured, fractal metrics instead reflect the underlying symmetry of spatial scale present in cloud fields and the physics governing  Earth's atmosphere more broadly.

We've also added that the resolution of the MODIS data we use is 1 km in response to a below comment.

*"L26: The two papers cited here are pretty clear about the measure of uncertainty, so the authors could be more clear about the struggle to accurately represent the radiative effects of clouds. Also, there have been quite a number of studies recently on this very topic since 2020."*

We rephrased the sentence to be more specific and added three references to studies published 2020 and after.

>  Clouds are also a particularly challenging test (Stephens et al., 2015) (Ceppi et al., 2017; Sherwood et al., 2020), as feedbacks between cloud radiative effects and surface temperature remain the most uncertain component of model-derived estimates of the climate sensitivity (Stephens et al., 2015; Ceppi et al., 2017; Zelinka et al., 2020; Sherwood et al., 2020; Arias et al., 2021; Bock and Lauer, 2024).

*"L87: It would be nice to see the actual resolution, and exactly how much it varies."*

Thank you for noticing this omission. We added:

To analyze the geometries of cloud fields, we use a calibrated optical reflectance product $R$ from MODIS that is sensitive to visible wavelengths between 620 nm and 670 nm (Band 1). The dataset has a nadir resolution of 1 km, increasing to roughly 2 km at a sensor zenith angle of 60°. We only consider the portion of the swath that has a sensor zenith angle of 60° or less.

*"L120 A "simple linear fit" can mean two things. Either that the (unwieldy) distribution has been estimated by a linear function, or that the power-law fit worked exceptionally well. Can we have a figure that actually shows the linear fit? "*

As described in section 3.2 (Measurements), we provide Figs. 5 and 6 showing the fit. Also, there is a linear relationship between $\log p$ and $\log \sqrt{a}$, which implies a power-law relationship between $p$ and $a$, so we are unsure of the "two things" to which the reviewer refers. We also note that this section does not discuss size distributions, although size distributions are mentioned elsewhere, and we are unsure if the reviewer meant size distribution by the word "distribution" or simply a relationship between areas and perimeters. More broadly, the point we are making in this section is that the simple fitting procedure the reviewer is referencing is often done in past studies but is *not* valid for clouds. We then go on to propose a different method, fitting $\log p$ to $\log \sqrt{a}$ after filling cloud holes, which improves the power-law behavior as shown in Fig. 5.

*"L214 It is indeed true that the perimeter and the size of a cloud are positively correlated, but the relationship is not linear (cf. https://doi.org/10.5194/acp-13-7795-2013). Either this significantly affects the power-law relationship, or I think it should be made clear that the power-law fit is not an exact representation of the cloud size distribution. "*

We apologize if we did not understand the question; however, around L214 in the original manuscript we are proposing a relationship between perimeter and *number*, not size, and furthermore we propose a power-law relationship, not a linear one. For another comment, we also added Fig 12, which was previously in the appendix and shows that a power law perimeter distribution is indeed a close fit to the observed perimeter distribution.
If the reviewer meant to reference the discussion around Figs. 5 and 6, we again find a power-law relationship between perimeter and area, not a linear one, as discussed in the previous comment. In fact, the relationship found in Fig. 6 appears consistent with the $p \propto a^{0.73}$ dependence mentioned in the abstract of the paper the reviewer linked.

*"L225 I believe the authors are being intentionally ambiguous here, as the cloud field geometry being "tied to" turbulent processes can be used to mean anything; if the authors are implying that the ensemble fractal dimension $D_e$ from satellite images can be used to infer the strength of turbulent mixing processes, or thermodynamic properties, then I would love to see more details. "*

That was not our intention. Rather, the manuscript is intended to serve as a methodological paper describing how fractal metrics may be measured, not what their physical interpretation in terms of turbulent mixing may be. Nonetheless, other studies have proposed such physical links, and this motivates why we might want more accurate methods of measuring fractal metrics in the first place. We have made a small change to indicate that these physical links are expanded upon in other past work:

If true, the ensemble fractal dimension given by Eqn. 6 provides a simple observational metric for quantifying cloud field geometries, one that  has previously been related to the invisible turbulent processes that roughen each individual cloud's edge ($D_i$; Hentschel and Procaccia, 1984; Siebesma and Jonker, 2000) as well as the physics controlling the competition for available energy among clouds of varying sizes ($\beta$; Garrett et al., 2018).

*"L280 There is very little discussion on how the linear fits in Figure 10 were obtained. The power-law distributions on the log-log plots are only marginally linear, and they are dependent on the choice of the bins and other factors (see the comment below); one could also argue that the intermediate sizes used for the linear fit have been cherry-picked. There is a lot of literature on using the power-law fit for the cloud size distribution, and I believe there is much to be justified here. "*

To address the fitting point, we added the following:

$$N(\varepsilon) \propto \varepsilon^{-D_{\mathrm{box}}}. \tag{1}$$

From this equation, $D_{\mathrm{box}}$ may be estimated using a least-squares linear regression to a plot of $N(\varepsilon)$ vs. $\varepsilon$. Note that the box dimension is equal to the usual dimension for  Euclidean objects; for example, a line has $D_{\mathrm{box}} = 1$ and a disk has $D_{\mathrm{box}} = 2$ (Strogatz, 2018).

For the rest of the comment, Figure 10 is a plot of the box-counting function $N(\varepsilon)$ as a function of box size, not a distribution of cloud sizes, so there is no binning done here. We agree that the intermediate sizes are somewhat "cherry-picked" in this case. We ultimately recommend that future work does *not* use this method for the reason the reviewer notes (in addition to the argument at line 274 in the revised manuscript), and we describe the method because it is currently more common than the correlation dimension. Thus, this section is at least partly attempting to make the reviewer's point: that the box dimension is not the best measure of cloud scaling properties. We added the following to clarify:

These values are somewhat higher than those obtained using the box dimension but display a similar trend of decreasing values with increasing $R$. As shown in Fig. 11, the correlation integral $C(r)$ is better represented by a power-law when compared to the box-counting function $N(\varepsilon)$ (Fig. 10), implying estimates of $D_c$ are more accurate when compared to $D_{\mathrm{box}}$.

It seems the reviewer may have been expecting to see a distribution plot of cloud perimeters. We have added one, Fig. 12, and figure numbering has been updated accordingly.

*"L281 This sounds like a significant amount of large clouds being filtered out. I do understand the need to filter truncated clouds, but given that larger clouds naturally have higher chances of extending beyond the observed domain, it sounds like a considerable chunk of the cloud field that is being omitted. For a typical MODIS scene, how much of the observed cloud field has been filtered out at this stage? The authors mention that including those clouds will result in "a substantial bias", but could it be the other way around? Also, why the bins that include 50% of whole clouds? These choices seem too arbitrary to me. Ultimately, I think this directly opposes the main goal of this paper, which is to come up with a measure of geometric features of a cloud field, especially for convective organization. "*

Thank you, the text may not have been sufficiently clear. The problem of accurately estimating $\beta$ is in itself a significant methodological challenge, and in a prior study we investigated the problem in detail using both numerical tools and observations of clouds. The 50% threshold, for example, was recommended there for future work after considering multiple options and the resulting bias that was observed for each method. Although it does omit a portion of the cloud field, it was found to be necessary because the perimeters of clouds that are truncated by the domain edge cannot be estimated. Here, we intended to mention the problem but defer to that prior study the details of the method. We have updated the following paragraph to emphasize that the method is described in full detail in the prior study:

To evaluate the hypothesis that $D_e = \beta D_i$ (Eqn. 6), $\beta$ is determined directly from the MODIS cloud masks.  Accurately determining the power-law distribution exponent in satellite data can be surprisingly challenging, primarily because the artificial truncation of clouds along the domain edge can influence the measured cloud size distribution. A comprehensive analysis of the possible biases arising from domain truncation was done by DeWitt and Garrett (2024). Here, we implement their recommended methodology, which was to calculate $\beta$ using a linear regression to a logarithmically binned and transformed histogram of cloud perimeter  (Fig. 12). As recommended by DeWitt and Garrett (2024), fits are performed only for those bins for which at least 50% of clouds in that bin are entirely contained within the measurement domain. This method ensures that $\beta$ is calculated from only the unbiased portion of the distribution, i.e. the portion that is not dominated by large clouds that extend beyond the measurement domain.

As for "how much of the observed cloud field has been filtered out at this stage", we calculated the fraction of total perimeter and total number of clouds that are removed at the large end of the spectrum for two thresholds for illustration. By count, and for threshold $R = 0.05$, 0.4% of all clouds are removed, and for $R = 0.2$, 0.06% of all clouds are removed. These removed clouds account for 30% and 16% of the total cloud perimeter, respectively.

As for whether the methodology "opposes the main goal of the paper", we respectfully disagree. The main goal of the paper is to point out that subtle measurement biases may result when measuring parameters such as $\beta$ and the fractal dimensions if sufficient care is not taken. The truncation threshold is also necessary to ensure the domain geometry does not bias estimates of $\beta$. We believe that DeWitt and Garrett (2024) provides ample evidence that the bias is not "the other way around" and hope the text changes above make it more clear that we are relying on this past analysis.

*"Figure 10 The exponents are missing on the y-axis (same for Figure 11). I am not sure if I agree with the observation. The slope $D_{box}$ does tend towards 0 for small and large values of $\varepsilon$ (for very different reasons), but I am not sure if we can say it tends towards 1, unless the authors mean that the slope is close to 1 for small $\varepsilon$, in which case I disagree. "*

Thank you, the figure tick marks have been fixed. Since the slope of the plot is equal to $-D_{\text{box}}$, we do mean that the slope tends toward -1 for small $\varepsilon$, as can be seen in the smallest two points, which have a flatter slope than the portion to which the regression is done.

*"L300 I do not see how this number came about; is 1.7 just an average across all estimates of the ensemble fractal dimension? I feel like these numbers depend more on the filtering of large clouds and the linear fit than the actual properties of the cloud field, as they behave differently at smaller thresholds. "*

Yes, as shown in Table 1, the majority of the estimates for the ensemble fractal dimension are close to 1.7, with some exceptions noted in the text. We made this change:

 As shown in Table 1, all three methods for calculating the ensemble fractal dimension – the box dimension, correlation dimension, and $\beta D_i$ – point to a value of $D_e \approx 1.7$, with a possible exception of the box dimension at higher reflectivity thresholds.

We hope the specific concerns discussed above address the reviewer's point about filtering. If the comment refers to additional methodological concerns, we are unsure of what threshold (e.g. cloud perimeter, reflectivity, or something else) is being referred to, and request the comment be clarified if it has not already been addressed above.

*"L330 I believe it should be "strict definition of the fractal dimension". "*

Thank you for catching this.

> Not filling cloud holes violates the strict definition  fractal dimension, while filling the holes misses an important physical property defining the cloud field.

*"L332 See my first comment. "*

Please see our response to that comment and check whether the changes made are satisfactory.

**Response to Reviewer 2's Comments**

*"This paper discusses different definitions of fractal dimensions for cloud fields. Its detailed exploration of what different fractal dimensions actually mean, and how they behave across different cloud fields is a worthwhile addition to the literature, and it should be published. The paper is clearly written, although sometimes a bit long and could serve from a small additional round of editing (e.g., I'm not sure that it is necessary to cite the Feynman lectures here). "*

Thank you. We have removed the Feynman citation as suggested.

*"I am not convinced about the framing of this paper as an objective classification of cloud fields. For that to be the case, I would expect some kind of relevant difference to occur between clouds in different classes of D, beyond the relatively trivial relationship with other metrics. I could guess that there are certain distinct differences (e.g., certain values correspond roughly with cumulus, stratus, gravel, fish, etc – but in a more precise way), but the paper is not actually that case. I recommend either making that case (in which case the paper becomes likely quite different, for instance by applying De to Janssens' cloud botany), or reframing the introduction and conclusions to simply focus on fractal properties. "*

Thank you for raising this point, as it brings up a distinction that we hope to better clarify. To rephrase the reviewer's point as we understand it: it is one thing to propose a metric that can be used to infer something about the cloud field. It is another to propose that such a metric may be used to distinguish one meteorological state from another, because then we would additionally require that the value of the proposed metric varies based on the local atmospheric conditions.
As the reviewer notes, we do not currently test whether any of the metrics we consider vary with the local meteorological conditions, and consider this question beyond the scope of the present paper. However, we do believe that it is appropriate to suggest that future studies could, in principle, use our metrics for such classification studies. Furthermore, some of these metrics have already been used for classification, so there is a precedent for using them in this manner and we believe this is sufficient justification for the claim that these metrics could be used for classification in the future. We added some discussion of this prior work in response to the reviewer's other comment about L63 below (see changes there), which might also partly address this comment.
Regardless, any classification metric requires, as a prerequisite, accurate methods for its calculation. It is this point where we believe our study adds value. We have made several changes to emphasize these points, first in the introduction:

>  As an alternative to subjective categories, here we propose that cloud fields may be better characterized by their geometry given that cloud boundaries are shaped by physics.

In the conclusions:

> Calculation of the ensemble fractal dimension $D_e$ bypasses these issues and  may offer a more objective alternative to a subjective classification scheme such as sugar, gravel, fish, and flowers (Stevens et al., 2020), particularly if calculated as a correlation integral. The ease of its calculation  using the correlation integral in satellite imagery makes it well-suited for evaluating the accuracy of atmospheric numerical simulations or for comparing regional meteorology.

*"Around line 260, and further on around line 280, I would like the authors to be more specific about their methods (e.g., "ignoring the portion the distribution that is dominated by large clouds that extend beyond the measurement domain") , and the biases their choices may or may not introduce. For instance, in the method section it was stated that clouds over land were not included, but that could easily bias against larger clouds.Likewise, the exclusion on L 260 does something similar. "*

This point was also brought up by the other reviewer. We did not make it sufficiently clear that the methodology for calculating $\beta$ was already analyzed in detail in a previous paper, DeWitt and Garrett (2024). We have updated the relevant paragraph to emphasize that the methods, the reasoning behind them, and the bias that could result if such methods were not used are described in detail in that past paper:

> To evaluate the hypothesis that $D_e = \beta D_i$ (Eqn. 6), $\beta$ is determined directly from the MODIS cloud masks.  Accurately determining the power-law distribution exponent in satellite data can be surprisingly challenging, primarily because the artificial truncation of clouds along the domain edge can influence the measured cloud size distribution. A comprehensive analysis of the possible biases arising from domain truncation was done by DeWitt and Garrett (2024). Here, we implement their recommended methodology, which was to calculate $\beta$ using a linear regression to a logarithmically binned and transformed histogram of cloud perimeter  (Fig. 12). As recommended by DeWitt and Garrett (2024), fits are performed only for those bins for which at least 50% of clouds in that bin are entirely contained within the measurement domain. This method ensures that $\beta$ is calculated from only the unbiased portion of the distribution, i.e. the portion that is not dominated by large clouds that extend beyond the measurement domain.  DeWitt and Garrett (2024)

It is true that we do not consider clouds over land, because land makes a simple reflectance-based threshold such as that used here insufficient for identifying clouds. Although we have found in prior work (DeWitt et al., 2024) that there is little difference between land- and ocean- based cloud perimeter distributions, as the reviewer mentioned in the previous comment, investigation of such differences is not the main goal of the present paper.

Note that we have also added Fig. 12 showing the computed perimeter distributions are very close to power-law, which may address part of the comment relating to larger clouds. We find the frequency of large clouds is what would be expected from prior work on global size distributions (e.g. (DeWitt et al., 2024; Wood and Field, 2011)).

As for line 260, we changed the threshold as follows to be more consistent with the reasoning stated in the text:

> An important subtlety for calculation of $D_c$ is that, without proper care, even a few small circles might extend beyond the domain boundary. If $C_j(r)$ is computed including such circles, $D_c$ will be biased towards the dimension of the artificially straight domain boundary of $D_c = 1$. To remedy this issue, we  ensure that no circles

> extend beyond the domain boundary  by enforcing $r \leq r_{\max} = \min(L, W)/3$ where $W$ and $L$ are the domain width and length, respectively.

As described in the text, this procedure is necessary to prevent the finite domain size, and the clouds truncated by the domain boundary, from affecting the results. This has the potential to bias against larger clouds in the sense that we are unable to measure very large-scale statistics due to the finite domain size, but this is a limitation of any dataset.

*"L 63: Please give some citations "*

The relevant sentence has been expanded to describe prior work using fractal metrics:

> Not all of the metrics we discuss are novel, as various fractal dimensions and size distribution exponents have previously been used to classify individual cloud types such as cirrus and cumulus (Batista-Tomás et al., 2016), cloud field types such as sugar and gravel (Janssens et al., 2021), and compare simulated and observed clouds (Siebesma and Jonker, 2000; Christensen and Driver, 2021; Raghunathan et al., 2025). But, as we show in Sect. 3 there are important pitfalls to fractal dimension analyses that have been almost entirely overlooked in prior literature. Specifically, the fractal dimension of clouds is  most commonly determined from a relationship between individual cloud perimeter and area, as first introduced by Lovejoy (1982).

*"L78: What is the pixelsize/resolution of the satellite images? "*

Thank you for noticing this omission. We added:

> To analyze the geometries of cloud fields, we use a calibrated optical reflectance product $R$ from MODIS that is sensitive to visible wavelengths between 620 nm and 670 nm (Band 1). The dataset has a nadir resolution of 1 km, increasing to roughly 2 km at a sensor zenith angle of 60°. We only consider the portion of the swath that has a sensor zenith angle of 60° or less.

*"L86: 72 images is not a whole lot to process into powerlaws. Is it possible to generate a larger dataset? What is the resulting number of clouds that are analyzed, and the margin of error in some of the later analysis as a result? "*

This is a good point. We have added a Supplement containing a sensitivity analysis and this note in the Datasets section:

> As shown in the Supplement, computed parameter values can vary for individual images but converge when computed over roughly 10-15 randomly selected images. This indicates that the parameter values computed here, obtained using all 72 images, represent their climatological values.

The text in the Supplement and the associated figure are shown below.

> Here we compute the parameters $\beta$, $D_i$, $D_c$, and $D_{\mathrm{box}}$ for varying dataset sizes. We consider reflectance thresholds $R = 0.1$, $R = 0.2$, and $R = 0.3$. Without replacement, a varying number of images are randomly selected from the ensemble of 72 images and each parameter is calculated for the subset as described in the text. All thresholds and methodological considerations are identical to those used for the full dataset in the main text. As can be seen in Fig. 1, when the parameters are computed

using only one or two images, their values can differ by as much as 0.3, although the individual fractal dimension shows less dependence. Once the dataset is composed of at least 10 to 15 images, the resulting parameters converge to a value close to that obtained using all 72 images. This suggests that our full dataset of 72 images is large enough that statistical uncertainty arising from the finite sample size is negligible.

**References**

Batista-Tomás, A., Díaz, O., Batista-Leyva, A., and Altshuler, E.: Classification and dynamics of tropical clouds by their fractal dimension, Quarterly Journal of the Royal Meteorological Society, 142, 983–988, 2016.

Ceppi, P., Brient, F., Zelinka, M. D., and Hartmann, D. L.: Cloud feedback mechanisms and their representation in global climate models, WIREs Climate Change, 8, e465, https://doi.org/https://doi.org/10.1002/wcc.465, 2017.

Christensen, H. M. and Driver, O. G. A.: The Fractal Nature of Clouds in Global Storm-Resolving Models, Geophysical Research Letters, 48, e2021GL095 746, https://doi.org/https://doi.org/10.1029/2021GL095746, 2021.

DeWitt, T. D. and Garrett, T. J.: Finite domains cause bias in measured and modeled distributions of cloud sizes, Atmospheric Chemistry and Physics, 24, 8457–8472, https://doi.org/10.5194/acp-24-8457-2024, 2024.

DeWitt, T. D., Garrett, T. J., Rees, K. N., Bois, C., Krueger, S. K., and Ferlay, N.: Climatologically invariant scale invariance seen in distributions of cloud horizontal sizes, Atmospheric Chemistry and Physics, 24, 109–122, https://doi.org/10.5194/acp-24-109-2024, 2024.

Garrett, T. J., Glenn, I. B., and Krueger, S. K.: Thermodynamic constraints on the size distributions of tropical clouds, Journal of Geophysical Research: Atmospheres, 123, 8832–8849, 2018.

Hentschel, H. G. E. and Procaccia, I.: Relative diffusion in turbulent media: The fractal dimension of clouds, Phys. Rev. A, 29, 1461–1470, https://doi.org/10.1103/PhysRevA.29.1461, 1984.

Janssens, M., Vilà-Guerau de Arellano, J., Scheffer, M., Antonissen, C., Siebesma, A. P., and Glassmeier, F.: Cloud Patterns in the Trades Have Four Interpretable Dimensions, Geophysical Research Letters, 48, e2020GL091 001, https://doi.org/https://doi.org/10.1029/2020GL091001, 2021.

Lovejoy, S.: Area-perimeter relation for rain and cloud areas, Science, 216, 185–187, 1982.

Raghunathan, G. N., Blossey, P., Boeing, S., Denby, L., Ghazayel, S., Heus, T., Kazil, J., and Neggers, R.: Flower-Type Organized Trade-Wind Cumulus: A Multi-Day Lagrangian Large Eddy Simulation Intercomparison Study, Journal of Advances in Modeling Earth Systems, 17, e2024MS004 864, https://doi.org/https://doi.org/10.1029/2024MS004864, e2024MS004864 2024MS004864, 2025.

Sherwood, S. C., Webb, M. J., Annan, J. D., Armour, K. C., Forster, P. M., Hargreaves, J. C., Hegerl, G., Klein, S. A., Marvel, K. D., Rohling, E. J., Watanabe, M., Andrews, T., Braconnot, P., Bretherton, C. S., Foster, G. L., Hausfather, Z., von der Heydt, A. S., Knutti, R., Mauritsen, T., Norris, J. R., Proistosescu, C., Rugenstein, M., Schmidt, G. A., Tokarska, K. B., and Zelinka, M. D.: An Assessment of Earth's Climate Sensitivity Using Multiple Lines of Evidence, Reviews of Geophysics, 58, e2019RG000 678, https://doi.org/https://doi.org/10.1029/2019RG000678, 2020.

Siebesma, A. P. and Jonker, H. J. J.: Anomalous Scaling of Cumulus Cloud Boundaries, Phys. Rev. Lett., 85, 214–217, https://doi.org/10.1103/PhysRevLett.85.214, 2000.

[Figure]

Figure 1: Parameter values computed as in the main text but for randomly selected subsets of the MODIS dataset.

Stephens, G. L., O'Brien, D., Webster, P. J., Pilewski, P., Kato, S., and Li, J.-l.: The albedo of Earth, Reviews of Geophysics, 53, 141–163, https://doi.org/https://doi.org/10.1002/2014RG000449, 2015.

Stevens, B., Bony, S., Brogniez, H., Hentgen, L., Hohenegger, C., Kiemle, C., L'Ecuyer, T. S., Naumann, A. K., Schulz, H., Siebesma, P. A., Vial, J., Winker, D. M., and Zuidema, P.: Sugar, gravel, fish and flowers: Mesoscale cloud patterns in the trade winds, Quarterly Journal of the Royal Meteorological Society, 146, 141–152, https://doi.org/https://doi.org/10.1002/qj.3662, 2020.

Strogatz, S. H.: Nonlinear dynamics and chaos: with applications to physics, biology, chemistry, and engineering, CRC press, 2018.

Wood, R. and Field, P. R.: The distribution of cloud horizontal sizes, Journal of Climate, 24, 4800–4816, 2011.